# Learning-augmented Rent-or-Buy with a Sample

**Davidson Zhu** [1]   **Sreenivas Gollapudi** [2]   **Debmalya Panigrahi** [3]

## Abstract

In this paper, we study the rent-or-buy problem (also called the Bahncard problem) in the learning-augmented setting. In this problem, a traveler must complete a sequence of trips that are revealed online over time, each of which has an associated cost with it. The traveler has the option of buying a discount card at a fixed cost that gives a discount on trip costs for a fixed time after buying the card. The goal is to minimize the overall cost of all the trips, including the money spent on buying discount cards. For this problem, it is well-known that the best deterministic algorithm has a competitive ratio of 2. In this paper, we ask whether we can do better if the traveler has a *sample* of trips available offline, e.g., obtained from an ML model based on historical data. We show that even a sparse sample of the input can significantly improve the competitive ratio of the algorithm from 2 to $3/2$, and further to close to 1 under some additional conditions. We also verify our theoretical bounds via numerical simulations, which reveal that our proposed algorithm obtains nearly optimal solutions for a variety of natural input classes.

## 1. Introduction

Recently, learning-augmented online algorithms have emerged as a central area of study at the intersection of algorithms and machine learning (Mitzenmacher & Vassilvitskii, 2020; 2022). In the classical online setting, algorithms make decisions based solely on past information and are evaluated against worst-case future inputs. By augmenting online algorithms with machine-learned predictions about the future, a growing body of work has shown that it is often possible to significantly improve upon worst-case competitive guar-

antees. A prominent example of this line of work augments online algorithms with a sample of the entire input. This model has been successfully applied to a variety of problems including online matching, secretary problems, load balancing, and network design (Lattanzi et al., 2020; Kumar et al., 2019b; Kaplan et al., 2020; 2022; Argue et al., 2022). In these works, the algorithm has offline access to a random sample of the instance, while the remaining input arrives online. The goal is to leverage the sample to obtain competitive bounds that improve on the best worst-case algorithms for the respective problems.

In this paper, we study online algorithms with a sample for classical rent-or-buy problems. In the simplest setting, called ski rental (Borodin & El-Yaniv, 2005), the online algorithm has to choose between buying skis at a fixed one-time cost or renting for every ski trip at a smaller but recurring rental cost. The difficulty lies in the fact that the instance is online, i.e., the number of skiing trips is not known at the outset. A more general setting is the Bahncard problem (Fleischer, 2001), where the one-time decision framework of ski rental is generalized in multiple ways. Here, trips arrive sequentially over time, and buying a card no longer covers all future trips, but only those within a fixed time interval for which the card is valid. In addition, trips may have distinct cost, and a valid card reduces the cost of covered trip to a $\beta$-fraction of the original cost for some $\beta \in [0, 1]$.[1]

Both the ski rental and Bahncard problems have been extensively studied in the last few years in learning-augmented settings (Purohit et al., 2018; Gollapudi & Panigrahi, 2019; Wang et al., 2020; Wei & Zhang, 2020; Anand et al., 2020; Zhao et al., 2024). These works focus on designing better solutions when explicit predictors provide sophisticated information such as optimal dual parameters (Zhao et al., 2024). In sharp contrast, we simply assume that the algorithm has access to a random sample of the input, thereby avoiding the need for sophisticated predictors. Formally, our setting is as follows. Let $\mathcal{I}$ be a ski rental or Bahncard instance. For a fixed sampling rate $0 < \varepsilon \leq 1$, every trip in $\mathcal{I}$ is independently included in the sample $\hat{\mathcal{I}}$ with proba-

[1]Duke University, Durham, NC, USA [2]Google Research, Mountain View, CA, USA [3]Department of Computer Science, Duke University, Durham, NC, USA. Correspondence to: Davidson Zhu <davidsonjiaduo.zhu@duke.edu>.

*Proceedings of the 43rd International Conference on Machine Learning*, Seoul, South Korea. PMLR 306, 2026. Copyright 2026 by the author(s).

---

[1]It often suffices to consider $\beta = 0$, since for positive $\beta$, the algorithm always pays a fixed $\beta$-fraction of each trip cost, and the decisions of the online algorithm to buy cards only affect the remaining $(1 - \beta)$-fraction.

bility $\varepsilon$. The algorithm sees $\hat{\mathcal{I}}$ before any trip arrives, and then processes the full instance online using the sample as auxiliary information.

## 1.1. Our Contributions

Our main contribution in this paper are two new sample-augmented online algorithms for the Bahncard problem: Algorithms 5.2 and 5.3. Let $c$ denote the ratio between the maximum trip cost and the cost of buying a card. As $c \to 0$, we show that Algorithm 5.2 achieves a competitive ratio converging to $3/2$. This already improves on the worst-case setting, where the best (deterministic) algorithm has a competitive ratio of 2 (Fleischer, 2001). We further conditionally improve the bound in Algorithm 5.3: if each trip cost lies in the range $[\alpha c, c]$ for some constant $\alpha > 0$, then we show that the competitive ratio of Algorithm 5.3 converges to 1, i.e., it is nearly optimal.

As a warm-up for our main results on the Bahncard problem, we first consider the simpler special case of the ski rental problem. For this problem, we show that the naïve algorithm of computing the optimal solution on the (suitably rescaled) sample and applying this solution to the online instance already achieves near-optimal cost. In contrast, it turns out that this simple approach is fundamentally incorrect for the more general Bahncard setting: for the latter problem, we demonstrate that there are instances where the naïve algorithm is arbitrarily worse than the optimal solution. This justifies the design of the more sophisticated algorithms (Algorithms 5.2 and 5.3) that avoid the pitfalls of the naïve algorithm for this problem.

We complement our theoretical analysis with numerical simulations. Our empirical results confirm the advantages of Algorithms 5.2 and 5.3 across a range of natural input distributions and sampling regimes, and illustrate their robustness to noise in the sample. The experiments also show that the failure modes of the naïve sampling approach arise frequently in practically motivated settings, and not just on adversarially constructed instances.

Our results demonstrate that even a sparse random sample can be leveraged to significantly outperform the classical worst-case guarantees for rent-or-buy problems. This gives strong evidence that even without using sophisticated predictors, there is substantial benefit to be obtained from a learning-augmented approach for this foundational class of online problems.

## 2. Related Work

**Rent-or-Buy Problems.** Rent-or-Buy is a popular class of online problems where the algorithm has to decide between a larger upfront cost (the "buy" option) and smaller recurring costs (the "rent" option), without knowing how many rental costs it will incur. The simplest setting is ski rental (Borodin & El-Yaniv, 2005), where buying skis once covers all rental costs of the future. The Bahncard problem considers a more general setting where buying a card gives a temporary discount reducing trip costs to a fraction $\beta$ of their original cost, until the card becomes invalid after a fixed length of time. This problem was introduced by Fleischer (Fleischer, 2001), who gave a (tight) $(2 - \beta)$-competitive deterministic algorithm and conjectured the existence of a (tight) $e/(e - 1 + \beta)$-competitive randomized algorithm for the problem; this conjecture was settled positively by (Karlin et al., 2003). Several variants of the Bahncard problem have since been studied, such as the multi-kind Bahncard problem (Timm & Storandt, 2020), Bahncard problem with interest rate and risk (Ding et al., 2005), and Bahncard problem with fluctuating price (Ding et al., 2009).

**Rent-or-Buy with Predictions.** There is a substantial line of work on the ski-rental problem with predictions. The problem was first studied in (Purohit et al., 2018). Subsequently, (Gollapudi & Panigrahi, 2019) investigated ski rental in a setting where predictions are provided in the form of expert advice, and (Shen et al., 2025) showed how calibrated advice can further improve performance in high-variance settings. In the opposite direction, (Wei & Zhang, 2020) established lower bounds demonstrating that certain consistency–robustness trade-offs are unavoidable for ski rental. For variants and generalizations of ski rental, (Wang et al., 2020) and (Shin et al., 2023) studied learning-augmented algorithms for multi-shop and multi-option ski-rental problems.

For the Bahncard problem, which generalizes ski rental, learning-augmented algorithms were proposed in (Bamas et al., 2020; Drygala et al., 2023; Zhao et al., 2024). The algorithm in (Bamas et al., 2020) is based on the primal-dual method for the online set cover problem, and its robustness–consistency guarantees require the card cost to grow unbounded. In a different direction, (Drygala et al., 2023) investigated how many perfect predictions are needed for a Bahncard algorithm to achieve near-optimal performance. The algorithm in (Zhao et al., 2024) generalizes the classical $(2\text{-}\beta)$-competitive algorithm and achieves improved consistency when the discount factor $\beta$ is large, while maintaining certain robustness guarantees.

In contrast to these works that use sophisticated predictions, we explore rent-or-buy algorithms that are simply provided a random sample of the input instance. Conceptually, our work an be thought of as evidence that even very simple (and sparse) predictions about the future can lead to significant performance gains for this important class of online algorithms.

**Online Algorithms with a Sample.** Prior to our work, online algorithms with a sample have been widely studied, but not for rent-or-buy problems. Introduced by Kaplan *et al.* (Kaplan et al., 2020; 2022), this framework allows the online algorithm offline access to a $p$-fraction of the instance selected uniformly at random; the remaining input is presented online in an adversarial order that may depend on the sample. Closely related models were also proposed by Kumar *et al.* (Kumar et al., 2019a) and further explored by Lattanzi *et al.* (Lattanzi et al., 2021). This framework has since been applied to a variety of problems, including load balancing, network design, facility location, and clustering, among others (Argue et al., 2022). A related line of work studies the random-order model, in which the portion of the instance not included in the sample arrives in a uniformly random order rather than an adversarial one. (Gupta & Singla, 2020) provides a comprehensive overview of random-order online models and their applications.

## 3. Problem Definition

A Bahncard instance $\mathcal{I}$ is given by a sequence of trips, each defined by the time at which the trip occurs and the *full* cost incurred. For the $i$th trip, we denote these by $t_i$ and $c_i$ respectively. The occurence of the $i$th trip and its cost are revealed online at time $t_i$. The algorithm can buy a card at any time $t$ at a fixed cost of $C$. A card purchased at time $t$ is valid for a duration $\Delta$, that is, over the interval $[t, t + \Delta]$. For any trip $i$, the algorithm pays the full cost $c_i$ if it does not have a valid card at time $t_i$, and pays the discounted cost $\beta c_i$ otherwise. Here, $\beta \in [0, 1]$ is a fixed discount factor that is common across all trips.

For notational simplicity, and without loss of generality, we normalize the length of time for which a card is valid and the cost of a card to unit values by setting $\Delta := 1$ and $C := 1$. We can then represent an instance $\mathcal{I}$ simply as the multi-set of its trips

$$\mathcal{I} = \{(c_i, t_i) : i \in [n], c_i, t_i > 0\},$$

where $n$ denotes the total number of trips. We assume that the trips are indexed in non-decreasing order of time, i.e., $t_i \leq t_j$ for all $i < j$. We also denote by $T$ the length of the entire instance, so that $t_i \leq T$ for all $i \in [n]$. Let $c := \max_{1 \leq i \leq n} c_i$ denote the maximum trip cost. Since the algorithm would always prefer to buy a card upon seeing any trip with $c_i \geq 1$, we assume w.l.o.g.[2] that $c \leq 1$.

**A Sampled Instance.** A sample $\hat{\mathcal{I}}$ for an instance $\mathcal{I}$ is a random subset of the trips in $\mathcal{I}$, obtained as follows. Each trip $(c_i, t_i) \in \mathcal{I}$ is independently included in $\hat{\mathcal{I}}$ with probability $\varepsilon$, where $\varepsilon \in (0, 1]$ is a fixed sampling rate. The

---

[2]without loss of generality

---

sampled instance is drawn from this distribution and is revealed to the algorithm before the online process begins.

For notational convenience, we rescale the cost of each sampled trip by a factor $\varepsilon^{-1}$. Let $A \subseteq [n]$ denote the indices of sampled trips. The sample can therefore be written as

$$\hat{\mathcal{I}} = \left\{ \left(c_i \cdot \varepsilon^{-1}, t_i\right) : i \in A \right\}.$$

**A Span on an Instance.** A span $P$ on instance $\mathcal{I}$ is a sub-instance consisting of a contiguous block of trips, namely

$$P = \{(c_i, t_i) : i \in [n_1, n_2]\} \subset \mathcal{I},$$

for some $1 \leq n_1 \leq n_2 \leq n$. The length of a span $P$ is defined as the length of the largest time interval containing exactly the trips in $P$, namely $\ell(P) := t_{n_2+1} - t_{n_1-1}$, where we set $t_0 = 0$ and $t_{n+1} = T$.

Conversely, any time interval $[t, t'] \subseteq [0, T]$ determines a span in the instance $\mathcal{I}$ consisting exactly of the trips $(c_i, t_i)$ with $t_i \in [t, t']$. We denote this span by $\mathcal{I}([t, t'])$. Note that different intervals may correspond to the same span. We say that a span $P$ (or its corresponding interval) starts at a trip $(c_i, t_i)$ if $(c_i, t_i)$ is the earliest trip in $P$.

**Full Cost.** Let $D \subseteq [n]$ and let $K = \{(c_i, t_i) : i \in D\} \subseteq \mathcal{I}$ be a subset of trips. On the original instance $\mathcal{I}$, define the total full cost of trips in $K$ by

$$c_{\mathcal{I}}(K) := \sum_{i \in D} c_i.$$

On the sampled instance $\hat{\mathcal{I}}$, denote the total full cost of trips in $K$ by

$$\hat{c}_I(K) := \sum_{i \in D \cap A} c_i \cdot \varepsilon^{-1}.$$

When the underlying instance is clear from context, we simply write $c(K)$ and $\hat{c}(K)$ for $c_{\mathcal{I}}(K)$ and $\hat{c}_{\mathcal{I}}(K)$, respectively. For a time interval $[t, t']$, we define its total full cost by $c_{\mathcal{I}}([t, t']) := c_{\mathcal{I}}(\mathcal{I}([t, t']))$ and $\hat{c}_{\mathcal{I}}([t, t']) := \hat{c}_{\mathcal{I}}(\mathcal{I}([t, t']))$.

**A Solution.** A Bahncard solution $\mathcal{S}$ is specified by the times at which cards are purchased. Accordingly, we represent a solution $\mathcal{S}$ as the set of purchase times

$$\mathcal{S} = \{b_j : j \in [m], b_j \geq 0\},$$

where $m$ is the number of cards, and each $b_j$ indicates that a card is bought at time $b_j$, providing a discount for all trips occurring in the interval $[b_j, b_j + 1]$.

**Cost of a Solution.** Fix an instance $\mathcal{I}$ and a solution $\mathcal{S}$. A trip $(c_i, t_i) \in \mathcal{I}$ is called *covered* if there exists some $b_j \in \mathcal{S}$

such that $t_i \in [b_j, b_j + 1]$, and *uncovered* otherwise. Let $L$ denote the set of covered trips.

The rental cost of $\mathcal{S}$ on $\mathcal{I}$ is defined as the total discounted cost of the covered trips plus the total full cost of the uncovered trips. We denote the rental cost by $\mathsf{rent}(\mathcal{S}, \mathcal{I})$, and have

$$\mathsf{rent}(\mathcal{S}, \mathcal{I}) = \beta \cdot c(L) + c(\mathcal{I} \setminus L).$$

The buy cost is defined as the total number of cards purchased in the solution $\mathcal{S}$, and is denoted by $\mathsf{buy}(\mathcal{S})$.

The total cost of $\mathcal{S}$ on $\mathcal{I}$ is the sum of the rental cost and the buy cost, which we denote by

$$\mathsf{cost}(\mathcal{S}, \mathcal{I}) = \mathsf{buy}(\mathcal{S}) + \mathsf{rent}(\mathcal{S}, \mathcal{I}).$$

Let $\mathrm{OPT}(\mathcal{I})$ denote an optimal solution for instance $\mathcal{I}$, that is, a solution minimizing the total cost. Such a solution can be found offline in polynomial time as follows. Construct a directed graph whose vertices correspond to the trips. From each trip $(c_i, t_i)$, add two outgoing edges: one edge to $(c_{i+1}, t_{i+1})$ with weight $c_i$, and one edge to $(c_j, t_j)$, where $(c_j, t_j)$ is the earliest trip satisfying $t_j > t_i + 1$, with weight 1. Then any shortest path from the first trip to the last trip corresponds to an optimal solution $\mathrm{OPT}(\mathcal{I})$.

## 4. Warm-Up: Ski Rental with Samples

As a warm-up, we first consider the ski rental problem that represents a simpler special case of the Bahncard problem.

In the ski rental problem, the algorithm receives a sequence of trips online. Upon the arrival of each trip, the algorithm can either rent, paying a cost of $c$ to serve the current trip, or buy, paying a cost of 1 to cover all future trips at once.

The ski rental problem is a special case of the Bahncard problem in which

- the duration of a card $\Delta$ is the entire length of the instance $T$, i.e., $T = \Delta = 1$ (by scaling),
- the full costs of all trips are identical (denoted $c$), and
- the discount factor $\beta = 0$, i.e., trips are free once a card has been bought.

Under these assumptions, an instance is completely specified by a single parameter: the total number of trips, which we denote by $n$ as earlier.

A sample for the ski rental problem is defined in the same way as for the Bahncard problem: each trip is independently included in the sample with probability $\varepsilon$. Since only the total number of trips is relevant (and not their times), the sample can be viewed as a random variable $\hat{n}$ drawn from $\mathrm{Binom}(n, \varepsilon)$. The parameter $n$ is well-approximated by the sample, which gives rise to the following simple algorithm.

**Algorithm.** Find the optimal solution $\hat{\mathcal{S}} = \mathrm{OPT}(\hat{\mathcal{I}})$ offline and follow it on the original instance. Explicitly, the algorithm buys at time 0 if and only if $\hat{n} c \cdot \varepsilon^{-1} \geq 1$. Denote this algorithm and its cost by ALG.

**Analysis.** The algorithm above either buys a card at time 0 or never buys a card. Its decision is purely based on whether the sample $\hat{n}$ drawn from $\mathrm{Binom}(n, \varepsilon)$ is at least $\varepsilon c^{-1}$.

Intuitively, there are two possibilities, depending on the value of $nc$. When $nc$ is close to 1, both choices—buying or not buying at time 0—performs well. On the other hand, when $nc$ is far from 1, the optimal solution on the sample is likely to coincide with the optimal solution on the original instance. Therefore, by simple case analysis and elementary application of concentration inequalities A.2, we establish the following theorem.

**Theorem 4.1.** *Let the competitive ratio of* ALG *be* $\psi$*. Then, as* $c \to 0$*, we get* $\psi \to 1$*.*

The full proof of Theorem 4.1 and the quantitative bound can be found in Appendix B.

## 5. Bahncard with Sampling

### 5.1. Naïve Algorithm

Motivated by the ski rental algorithm, a natural approach in the Bahncard setting is to use the sampled instance to estimate the future and follow an offline optimal solution on the scaled sample. Specifically, given a sample obtained by independently including each trip with probability $\varepsilon$, one may scale the sample by a factor of $1/\varepsilon$, compute an offline optimal solution on the sampled instance, and then follow this solution on the original instance. We now show that this naïve strategy can in fact perform arbitrarily poorly, even when the fixed upper bound $c$ on trip costs is small. For simplicity, we restrict attention to the case $\beta = 0$.

We construct an instance $\mathcal{I}$ as follows. Fix $\varepsilon$ and $c$, and let $0 < \tau < 1$ be any constant. Set $T = 1 + 3\tau$. Place $N$ trips, each of full cost $c$, uniformly in the interval $[\tau, 1 + 2\tau]$, with the first trip at time $\tau$ and the $N$th trip at time $1 + 2\tau$, where $N$ is a large parameter to be specified later. Next, place $2^N$ trips uniformly in each of the intervals $[0, \tau]$ and $[1 + 2\tau, 1 + 3\tau]$, with each of these trips having full cost $4^{-N}$.

With high probability, the longest run of unsampled trips has length $\log_{1/\varepsilon} N + O(1)$ (Schilling, 1990). Let this run start at time $t_1$ and end at time $t_2$. Then the above algorithm buys cards covering the intervals $[t_1 - 1, t_1]$ and $[t_2, t_2 + 1]$, since this is optimal for the scaled sampled instance. On the original instance, the algorithm pays $2 + c \cdot (\log_{(1/\varepsilon)} N + O(1))$, since no trip in the longest run is covered. In contrast, the optimal solution on the original

instance pays 2. Moreover, one can check that no optimal solution on the scaled sampled instance achieves lower cost on the original instance. Therefore, taking $N \to \infty$ makes the competitive ratio arbitrarily bad.

A visual illustration is given in Figure 1.

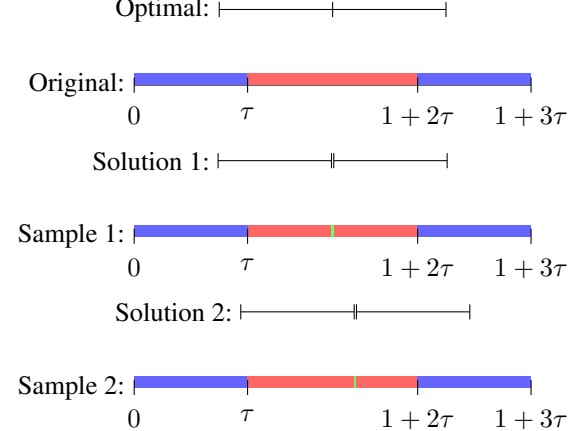

Figure 1. An illustration of the adversarial instance. The blue regions are densely populated with trips of negligible cost, while the red regions contain trips of cost $c$. With high probability, a sampled instance misses a consecutive run of trips (the small gap in green), in which case the naive algorithm chooses not to cover this interval because it yields only marginal benefit on the sample. This leads to a large additional cost on the original instance.

## 5.2. Algorithm 5.2: A $3/2$-competitive algorithm

From the adversarial construction above, we obtain the following intuition: a short gap containing few or no sampled trips may not actually be empty in the original instance. Instead, it may correspond to a region that is densely populated with trips which were simply missed during sampling. In Algorithm 5.2, we address this issue in the most direct way by eliminating all short gaps between cards by shifting card positions and, when necessary, inserting additional cards, thereby ensuring that no short gap remains between consecutive cards.

**Algorithm 5.2** Fix a parameter $0 < \delta < 1$, whose value will be specified later. Algorithm 5.2 first computes an offline optimal solution $\hat{S} = \text{OPT}(\hat{\mathcal{I}})$ on $\hat{\mathcal{I}}$.

In $\hat{S}$, if $[t, t+1]$ and $[t', t'+1]$ are two consecutive cards in $\hat{S}$, we call $g = (t+1, t')$ a *gap*. If $t' - (t+1) \geq \delta$, we call $g$ a *long gap*. Let $\{t_1, \ldots, t_k\}$ be the times where $\hat{S}$ buys cards, and let $L$ be the set of cards followed by long gaps. Then, the algorithm partitions the set of cards at these long gaps.

For each resulting group $\{t_{r+1}, \ldots, t_{r'}\}$, the algorithm discards all cards bought in $[t_{r+1}, t_{r'} + 1]$ and fully covers this interval using the minimum possible number of

---

**Algorithm 5.2**

1: **Offline Phase**
2: **Input:** (Scaled) sampled $\hat{\mathcal{I}}$, parameter $\delta \in (0, 1)$
3: Compute an offline optimum $\hat{S} \leftarrow \text{OPT}(\hat{\mathcal{I}})$
4: Let $t_1 < t_2 < \cdots < t_k$ be the buy times in $\hat{S}$
5: Let $L \leftarrow \{ r \in \{1, \ldots, k-1\} : t_{r+1} - (t_r + 1) \geq \delta \}$
   {Every $(t_r + 1, t_{r+1})$ is a long gap}
6: Initialize $S \leftarrow \varnothing$
7: Initialize $a \leftarrow 1$
8: **for** each $r \in L$ in increasing order **do**
9: $\quad b \leftarrow r$
10: $\quad s \leftarrow t_a$ {start time of current group}
11: $\quad e \leftarrow t_b + 1$ {end time of current group}
12: $\quad$ Add buy times $\{ s, s+1, \ldots, s + \lceil e - s \rceil - 1 \}$ to $S$
13: $\quad a \leftarrow r + 1$
14: **end for**
15: $s \leftarrow t_a$
16: $e \leftarrow t_k + 1$
17: Add buy times $\{ s, s+1, \ldots, s + \lceil e - s \rceil - 1 \}$ to $S$
18: **Online Phase**
19: Buy cards online on $\mathcal{I}$ according to $S$

---

cards, by buying cards at times $t_{r+1}, t_{r+1} + 1, \ldots, t_{r+1} + \lceil t_{r'} + 1 - t_{r+1} \rceil - 1$. The union of all these cards defines a new solution $S$. The online algorithm then buys cards according to $S$.

**Analysis.** The performance of Algorithm 5.2 is summarized by the following theorem.

**Theorem 5.1.** *By setting $\delta = c^{\frac{1}{5}}$, the competitive ratio of Algorithm 5.2 is bounded by $(3/2) \cdot (1 + \phi + c^{\frac{1}{5}})$, where $\lim_{c \to 0} \phi = 0$. In particular, Algorithm 5.2 is $(3/2)$-competitive as $c \to 0$. If $\varepsilon \to 0$ simultaneously, then Algorithm 5.2 is $(3/2)$-competitive if $(c/\varepsilon^5) \to 0$.*

Denote $S^* = \text{OPT}(\mathcal{I})$. To bound the competitive ratio of Algorithm 5.2, it suffices for us to bound $(\mathbb{E}[\text{cost}(S, \mathcal{I})] - b)/\text{cost}(S^*, \mathcal{I})$ for some universal constant $b$. To do this, we consider the following four factors:

$$\frac{\mathbb{E}[\text{cost}(S^*, \hat{\mathcal{I}})]}{\text{cost}(S^*, \mathcal{I})}, \frac{\text{cost}(\hat{S}, \hat{\mathcal{I}})}{\text{cost}(S^*, \hat{\mathcal{I}})}, \frac{\text{cost}(S, \hat{\mathcal{I}})}{\text{cost}(\hat{S}, \hat{\mathcal{I}})}, \frac{\mathbb{E}[\text{cost}(S, \mathcal{I})]}{\mathbb{E}[\text{cost}(S, \hat{\mathcal{I}})]}.$$

The first three factors are easy to bound and are handled together in the following lemma.

**Lemma 5.2.**

$$\mathbb{E}[\text{cost}(S, \hat{\mathcal{I}})] \leq \frac{3}{2} \cdot \text{cost}(S^*, \mathcal{I}).$$

*Proof.* Each uncovered trip $(c_i, t_i)$ is scaled by $1/\varepsilon$ and sampled with probability $\varepsilon$, so $\mathbb{E}[\text{rent}(S^*, \hat{\mathcal{I}})] = \text{rent}(S^*, \mathcal{I})$.

Therefore,

$$\mathbb{E}[\text{cost}(\mathcal{S}^*, \hat{\mathcal{I}})] = \text{cost}(\mathcal{S}^*, \mathcal{I}). \tag{1}$$

Since $\hat{\mathcal{S}}$ is optimal on $\hat{\mathcal{I}}$,

$$\text{cost}(\hat{\mathcal{S}}, \hat{\mathcal{I}}) \leq \text{cost}(\mathcal{S}^*, \hat{\mathcal{I}}). \tag{2}$$

By construction of $\mathcal{S}$, every trip in $\hat{\mathcal{I}}$ that is covered by $\hat{\mathcal{S}}$ is also covered by $\mathcal{S}$, so $\text{rent}(\mathcal{S}) \leq \text{rent}(\hat{\mathcal{S}})$. Consider an interval $[t'_i, t_j]$ between two long gaps. If $\hat{\mathcal{S}}$ buys $k$ cards, $t_j - t'_i < k + (k-1)\delta$ since all gaps in this interval are short. Then, to fully cover such an interval, $\mathcal{S}$ buys at most $k + \lceil (k-1)\delta \rceil$ cards. For $\delta < 1/2$,

$$\frac{k + \lceil (k-1)\delta \rceil}{k} \leq \frac{3}{2}.$$

Note that this bound is tight for arbitrarily small $\delta$ when $k = 2$. So, $\text{buy}(\mathcal{S}) \leq (3/2) \cdot \text{buy}(\hat{\mathcal{S}})$, which implies

$$\text{cost}(\mathcal{S}, \hat{\mathcal{I}}) \leq (3/2) \cdot \text{cost}(\hat{\mathcal{S}}, \hat{\mathcal{I}}). \tag{3}$$

Combining (1), (2), and (3) completes the proof. $\square$

We next need to bound $\mathbb{E}[\text{cost}(\mathcal{S}, \mathcal{I})]/\mathbb{E}[\text{cost}(\mathcal{S}, \hat{\mathcal{I}})]$, which is the main focus of the analysis.

**Lemma 5.3.** *By setting $\delta = c^{\frac{1}{5}}$,*

$$\frac{\mathbb{E}[\text{cost}(\mathcal{S}, \mathcal{I})]}{\mathbb{E}[\text{cost}(\mathcal{S}, \hat{\mathcal{I}})]} \leq 1 + \phi + c^{\frac{1}{5}},$$

*where $\lim_{c \to 0} \phi = 0$. In particular, Algorithm 5.2 is $(3/2)$-competitive as $c \to 0$. If $\varepsilon \to 0$ simultaneously, then Algorithm 5.2 is $(3/2)$-competitive if $(c/\varepsilon^5) \to 0$.*

*Proof of Theorem 5.1.* Immediately follows from Lemma 5.2 and Lemma 5.3. $\square$

The full proof for Lemma 5.3 is included in Appendix C. The quantitative bounds can be found in Lemma C.7, Lemma C.8, Lemma C.9, and Lemma 5.3. Here, we provide some high-level intuition for the main difficulty and our approach.

As in the ski-rental analysis, our proof relies on showing that, for the constructed solution, the total cost of uncovered trips in the original instance is not much larger than the expected uncovered cost in the sampled instance. This property holds trivially for $\mathcal{S}^*$ when we bounded $\mathbb{E}[\text{cost}(\mathcal{S}^*, \hat{\mathcal{I}})]/\text{cost}(\mathcal{S}^*, \mathcal{I})$, since $\mathcal{S}^*$ is fixed and sampling preserves expected cost.

The main difficulty is that our solution $\mathcal{S}$ strongly depends on the sample. In fact, the adversarial instance from Section 5.1 has demonstrated that the above cannot be bounded for our naïve algorithm. Our key idea is to exploit the structural property enforced by Algorithm 5.2—all uncovered gaps are long. This allows us to bound relaxed quantities with no dependency on $\mathcal{S}$.

## 5.3. Algorithm 5.3: Conditionally Improved Bounds

Algorithm 5.2 achieves a competitive ratio of $\frac{3}{2} \cdot (1 + \phi + c^{1/5})$. Recall that between two long gaps, if $\hat{\mathcal{S}}$ buys $k$ cards, $\mathcal{S}$ buys at most $k + \lceil (k-1)\delta \rceil$ cards. The factor of $3/2$ comes from the fact that even when there are only two cards with a very small gap in between, the algorithm still needs a card to cover it. As demonstrated by the adversary for the naïve algorithm, this obstruction is unavoidable under this algorithmic framework.

Nevertheless, note that this adversarial instance crucially needs trips of negligible cost. It turns out that if we disallow trips with very small cost and assume that each trip $(c_i, t_i)$ satisfies $\alpha c \leq c_i \leq c$ for some fixed constant $\alpha > 0$, we can get an algorithm with almost optimal asymptotic performance.

**Algorithm 5.3** The offline section of Algorithm 5.3 is similar to that of Algorithm 5.2, except that Algorithm 5.3 only covers the interval between two long gaps if there are at least $1/\delta$ cards bought in the interval. Otherwise, it merely aligns the right-most card to the last trip it covers.

Algorithm 5.3 in addition makes some modifications online: in an interval between two long gaps, when it encounters a gap with a significant amount of trips missing in the sample, the algorithm covers the rest of the interval.

**Analysis.** The performance of Algorithm 5.3 is summarized by the following theorem.

**Theorem 5.4.** *Assume that the cost of all trips are in the range $[\alpha c, c]$ for some constant $\alpha > 0$. By setting $\delta = c^{\frac{1}{5}}$, The competitive ratio of Algorithm 5.3 is bounded by $\max\{1 + \phi + c^{\frac{1}{5}}, 1 + 2c^{\frac{1}{5}} + (1 + c^{-\frac{1}{5}})(1 + c^{\frac{1}{5}})\phi'\}$, where $\lim_{c \to 0} \phi = 0$ and $\lim_{c \to 0}(1 + c^{-\frac{1}{5}})(1 + c^{\frac{1}{5}})\phi' = 0$. In particular, Algorithm 5.3 is 1-competitive as $c \to 0$. If $\varepsilon \to 0$ simultaneously, then Algorithm 5.2 is $(3/2)$-competitive if $(c/\varepsilon^5) \to 0$.*

The proof of Theorem 5.4 proceeds similarly to that of Theorem 5.1 with some extra technicalities, and we include the full proof in Appendix D. The quantitative bound can be found in Lemma D.6 and Lemma D.5.

## 6. Experiments

We conduct experiments comparing Algorithm 5.2 and Algorithm 5.3 with three baseline algorithms: the classical online algorithm without sampling (Counter), the naïve sampling-based algorithm from Section 5.1 (Naïve), and PFSUM (Zhao et al., 2024). For completeness, we briefly describe each algorithm below.

The classical algorithm Counter does not use the sample. It maintains a counter that tracks the accumulated rental cost.

**Algorithm 5.3**

1: **Offline Phase**
2: **Input:** (Scaled) sampled $\widehat{\mathcal{I}}$, parameter $\delta \in (0, 1)$
3: Compute an offline optimum $\widehat{S} \leftarrow \mathrm{OPT}(\widehat{\mathcal{I}})$
4: Let $t_1 < t_2 < \cdots < t_k$ be the buy times in $\widehat{S}$
5: Let $L \leftarrow \{\, r \in \{1, \ldots, k-1\} : t_{r+1} - (t_r + 1) \geq \delta \,\}$
   $\{$Every $(t_r + 1, t_{r+1})$ is a long gap$\}$
6: Initialize $\mathcal{S}_{\mathrm{off}} \leftarrow \varnothing$
7: Initialize $a \leftarrow 1$
8: **for** each $r \in L$ in increasing order **do**
9:    $b \leftarrow r$
10:   **if** $b - a + 1 \geq 1/\delta$ **then**
11:      $s \leftarrow t_a$ {start time of current group}
12:      $e \leftarrow t_b + 1$ {end time of current group}
13:      Add buy times $\{\, s, s+1, \ldots, s + \lceil e - s \rceil - 1 \,\}$ to $\mathcal{S}_{\mathrm{off}}$
14:   **else if** $b = a$ **then**
15:      Add buy time $\{t_a\}$ to $\mathcal{S}_{\mathrm{off}}$
16:   **else**
17:      Add buy times $\{t_a, t_{a+1}, \ldots, t_{b-1}\}$ to $\mathcal{S}_{\mathrm{off}}$
18:      Let $(c, t)$ be the last trip in $[t_b, t_b + 1]$
19:      Add buy time $\{t - 1\}$ to $\mathcal{S}_{\mathrm{off}}$
20:   **end if**
21:   $a \leftarrow r + 1$
22: **end for**
23: $s \leftarrow t_a$
24: $e \leftarrow t_k + 1$
25: Add buy times $\{\, s, s+1, \ldots, s + \lceil e - s \rceil - 1 \,\}$ to $\mathcal{S}_{\mathrm{off}}$
26: **Online Phase**
27: Initialize $a \leftarrow 1$
28: **for** each $r \in L$ in increasing order **do**
29:   $b \leftarrow r$
30:   Follow $\mathcal{S}_{\mathrm{off}}$ in $[t_a, t_b + 1]$ according to $\mathcal{S}_{\mathrm{off}}$, until on some gap $(t, t') \subset [t_a, t_b+1]$, $c(t, t'') \geq (1+\delta)/(1-\beta)$ for some $t'' < t'$, at which point the algorithm covers the rest of $[t_a, t_b + 1]$ using as few cards as possible.
31: **end for**
32: The solution realized online will be called $\mathcal{S}_{\mathrm{on}}$

Once this counter reaches $1/(1-\beta)$ (assuming the card cost is 1), the algorithm buys a card and resets the counter. This algorithm is well known to be $(2\text{-}\beta)$-competitive.

The algorithm (Naïve) uses the sampled instance by scaling it by a factor of $1/\varepsilon$, computing an optimal offline solution on the scaled sample, and then following this solution to determine when to buy cards.

The PFSUM algorithm buys a card at an uncovered trip at time $t$ if and only if both the total cost in the interval $(t-1, t]$ and the predicted cost in $[t, t+1)$ are at least $1/(1-\beta)$ (assuming that a card lasts one time unit and has a cost of

1). In our setting, the predicted cost in $[t, t+1)$ is estimated using the scaled sample. We have empirically verified that this provides the most accurate estimator among the natural choices.

**Input instances.** For all experiments, we fix the instance length to 20000 days and the card duration to 200 days. We evaluate performance under discount rates $\beta = 0, 0.2, 0.5, 0.7, 0.9$. We only consider arrival patterns in which at most one trip occurs per day. Allowing multiple trips per day improves concentration in the sample and strongly benefits sampling-based algorithms; since this regime is significantly easier, we focus on the more difficult single-arrival setting.

Following the experimental setup of (Zhao et al., 2024), we consider two arrival patterns: commuters, for which a trip arrives every single day, and occasional travelers, for which the time between two trips follow an exponential distribution. The commuter arrival models dense and regular arrivals, whereas the occasional arrival models more sparse and irregular arrivals. To capture large and small gaps, we consider exponential distributions with mean 2 and 40, respectively.

We additionally consider a third arrival pattern, which we refer to as clusters. In this model, trips arrive in bursts: whenever a trip occurs, it triggers a cluster of trips arriving together. The number of trips in a cluster follows a normal distribution with a mean of 50 and a variance of 25, and the gaps between clusters follow an exponential distribution of mean 500. Clustered or bursty arrival patterns have been observed in many real-world systems, such as transportation, web traffic and network requests. In these scenarios, trips concentrate around a few major events and are separated by longer idle periods (as modeled by exponential inter-arrival gaps).

Again following (Zhao et al., 2024), we consider two price distributions: a normal distribution with mean 50 and variance 25, and a Pareto (Lomax) distribution with shape parameter 2 and scale parameter 50, representing thin-tailed and heavy-tailed cost regimes, respectively. Since our algorithms assume that the card cost is large relative to individual trip costs, we cap all trip costs at 100: any sampled cost exceeding 100 is truncated to 100.

**Sampling.** For all input instances, we run the experiments with sampling rate 0.1 and 0.5. These two values are chosen to represent qualitatively different data regimes: a low-information regime, where the algorithm has access to only a small fraction of the instance, and a high-information regime, where the sample captures a substantial portion of the input. This allows us to evaluate the algorithms under both sparse and dense sampling.

Although we only analyzed the theoretical limit of our algorithms with a perfect sampling model, we run our algorithms on instances with noise to test its robustness. In practice, samples used to guide online decisions are rarely exact: they may be corrupted by measurement error, missing data, or distributional shifts between historical and future inputs. By adding controlled noise into the sampled instance, we evaluate the robustness of the algorithms to realistic perturbations, which is essential for deploying sampling-based methods in practical settings.

For experiments with noisy samples, we will set discount rate $\beta = 0$ and run the experiment for perturbation probability $p = 0, 0.1, 0.5, 0.9$, capturing regimes ranging from noiseless sampling to heavy corruption.

We follow the methodology of (Zhao et al., 2024) to create a noisy instance. Fix probability $p$, each travel request is independently removed from the sample with probability $p$, and independently, noise drawn from the underlying price distribution is added on each day with probability $p$. These two processes are independent. Then, the sample is created from the noisy instance.

**Results and discussion.** We only present the experimental result in the high-sampling regime without perturbation here and include the rest in Appendix E.

In every figure, each curve reports the average competitive ratio of an algorithm relative to the offline optimum, averaged over 100 independent runs. When the performance of the classical online algorithm without sampling or the naïve sampling-based algorithm is significantly worse than the others, we omit them from the plots for readability. The interval around each point represents a 95% confidence interval. Most confidence intervals are difficult to see because they are smaller than the marker size.

We first note that the experimental results demonstrate that our algorithms use the sample in a genuinely nontrivial way. In Section 5.1, we constructed a adversarial instance showing that Naïve can perform arbitrarily poorly. Here, we empirically confirm that even on natural input classes such as commuter arrivals, our algorithms have a clear performance advantage over Naïve. This is especially evident in the results presented in Appendix E.

We now focus on comparing our algorithms with PFSUM. Figure 2 show the performance of the algorithms under a high sampling rate ($\varepsilon = 0.5$).

In the high-sampling regime, our algorithms demonstrated high consistency across all arrival patterns and price distributions. On commuter instances and occasional instances, they achieve near-optimal performance comparable to that of PFSUM. On clustered instances, our algorithms significantly outperform PFSUM, highlighting their ability to

exploit sample information in highly structured inputs.

## 7. Closing Remarks

In this paper, we gave the first online algorithms for rent-or-buy problems that are augmented with a random sample of the input instance. We analyzed the algorithms theoretically as well as empirically, and concluded that our algorithms significantly outperform traditional online algorithms without predictions.

Our work suggests several potential future directions:

1. The competitive ratios of our algorithms depend on the $c$, the ratio of the maximum cost of a trip to the cost of a card. Is this dependence necessary?

2. Even when $c \to 0$, can we obtain an algorithm whose competitive ratio approaches 1 unconditionally?

3. Can our algorithms be adapted for other extensions of the ski rental problem with similar worst-case guarantees, such as the TCP acknowledgment problem (Dooly et al., 2001; 1998; Karlin et al., 2003)?

4. Is it possible to prove any performance lower bound for algorithms with sample? We remark that this seems especially challenging due to the interplay between randomness in the sample and the online decision-making process, which breaks standard adversarial constructions.

5. The algorithms developed in this paper are deterministic. Is it possible to obtain randomized algorithms with better performance guarantee?

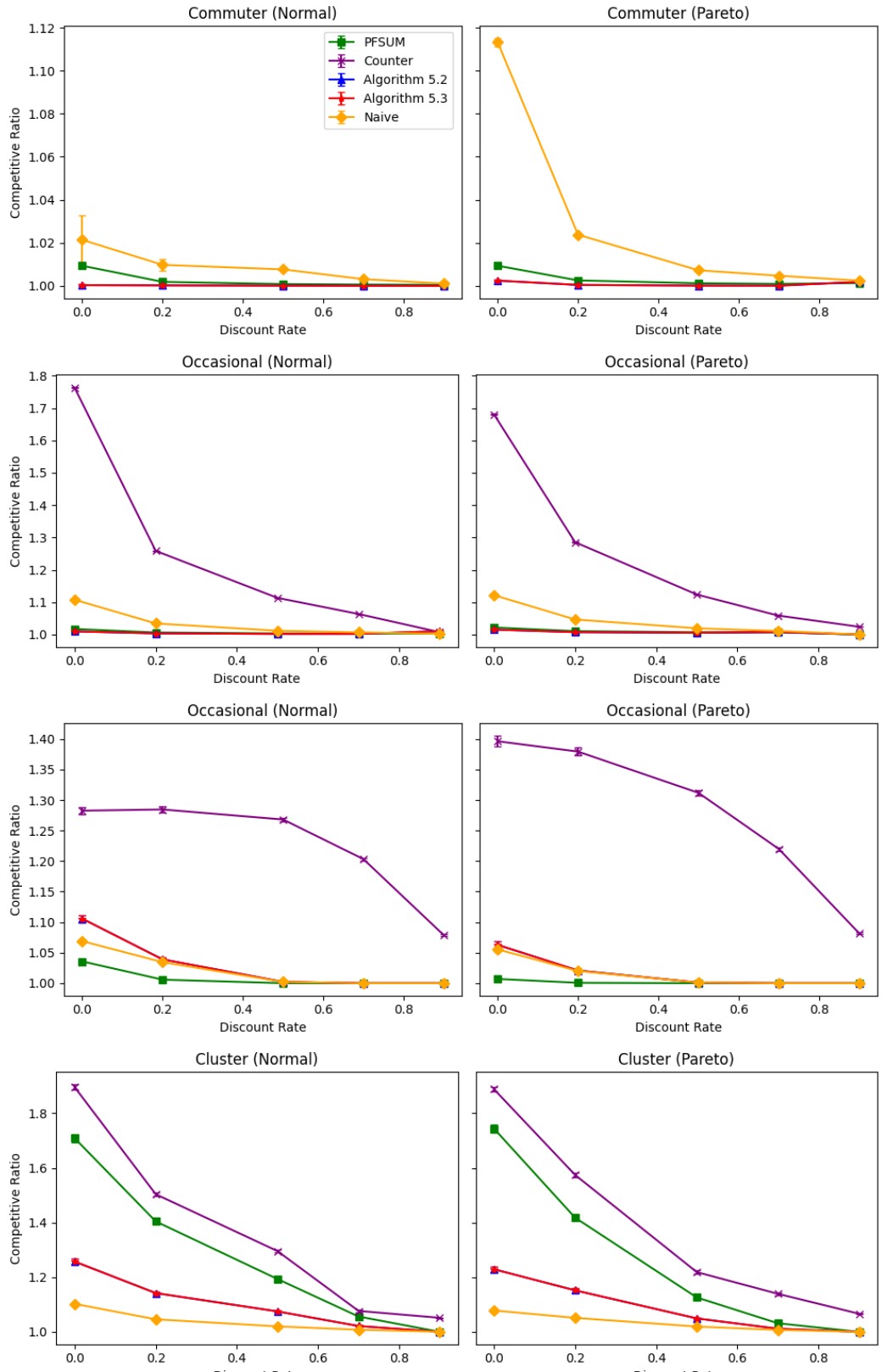

*Figure 2.* Cost ratios of the algorithms on instances with different trip arrival patterns, with sampling rate $\varepsilon = 0.5$. We compare PFSUM, Counter, Naïve, Algorithm 5.2, and Algorithm 5.3. The legend is shown in the first figure.

## Acknowledgments

Debmalya Panigrahi was supported in part by NSF grants CCF-2329230 and CCF-1955703. Part of this work was done when he was a Visiting Faculty Researcher at Google Research.

## Impact Statement

This paper presents work whose goal is to advance the field of Machine Learning. The work is theoretical in nature and no societal impact is anticipated.

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

# A. Standard Results

### A.1. Hoeffding's Bound

We use the following version of Hoeffding's inequality in our proofs:

**Theorem A.1** (Hoeffding's inequality (Hoeffding, 1963)). *Let $X_1, \ldots, X_n$ be independent random variables such that $a_i \leq X_i \leq b_i$. Let $S_n = X_1 + \cdots X_n$. Then, for all $t > 0$,*

$$\mathbb{P}\left(|S_n - \mathbb{E}[S_n]| \geq t\right) \leq 2 \exp\left\{-\frac{2t^2}{\sum_{i=1}^n (b_i - a_i)^2}\right\}.$$

We also use the following version of Chernoff bound for simplicity, which is a special case of Hoeffding's bound.

**Theorem A.2** (Chernoff bound(Mitzenmacher & Upfal, 2017)). *Let $X_1, \ldots, X_n$ be independent Bernoulli random variables (with possibly different distributions). Let $X$ be their sum and $\mu := \mathbb{E}[X]$ the expected sum. Then for any $\delta > 0$,*

$$\Pr[X \geq (1 + \delta)\mu] \leq \exp\{-\delta^2\mu/(2 + \delta)\}, \Pr[X \leq (1 - \delta)\mu] \qquad \leq \exp\{-\delta^2\mu/2\}.$$

# B. Proof of Theorem 4.1

By definition of ALG,

$$\mathbb{E}[\mathrm{ALG}] = \mathbb{P}\left[\mathrm{Binom}(n, \varepsilon) < \varepsilon \cdot c^{-1}\right] \cdot (nc) + \mathbb{P}\left[\mathrm{Binom}(n, \varepsilon) \geq \varepsilon \cdot c^{-1}\right] \cdot 1, \tag{4}$$

where $\mathrm{Binom}(n, \varepsilon)$ is the Binomial distribution.

We perform case analysis depending on the value of $nc$. When $nc$ is close to 1, we can bound the performance of ALG just by the fact that it can only buy a card at time 0. On the other hand, when $nc$ is far from 1, the probability that ALG makes non-optimal decision can be bounded by Chernoff bound (Theorem A.2).

Let $0 < \delta < 1$ be a parameter specifying what it means for $nc$ to be "close to 1".

**Case 1:** When $(1 + \delta)^{-1} \leq nc \leq 1$, ALG achieves optimum if it never buys, and the competitive ratio is $1/(nc) \leq 1 + \delta$ if ALG buys at time 0. Therefore, the competitive ratio of ALG is bounded by $1 + \delta$. Similarly, when $1 \leq nc \leq (1 - \delta)^{-1}$, the competitive ratio of ALG is bounded by $(1 - \delta)^{-1}$.

**Case 2:** If $nc < \varepsilon$, then $\hat{n}c \cdot \varepsilon^{-1} \leq 1$ since $\hat{n} \leq n$. Therefore, ALG never buys and is always optimal.

We are left with two remaining cases: when $nc$ is significantly greater than 1, and when $nc$ is significantly smaller than 1.

**Case 3:** When $nc \geq (1 - \delta)^{-1}$, we have that $\mu := \mathbb{E}[\mathrm{Binom}(n, \varepsilon)] = n \cdot \varepsilon \geq \varepsilon \cdot (c(1 - \delta))^{-1}$. Therefore, by Theorem A.2, we have

$$\mathbb{P}\left[\mathrm{Binom}(n, \varepsilon) < \varepsilon \cdot c^{-1}\right] \leq \mathbb{P}\left[\mathrm{Binom}(n, \varepsilon) < (1 - \delta) \cdot \mu\right] < \exp\left\{-\delta^2\mu/2\right\} < \exp\left\{-\delta^2 n\varepsilon/2\right\}.$$

Since the optimal cost is 1, by (4), the competitive ratio in this case is

$$\psi_1 := \mathbb{E}[\mathrm{ALG}] < \exp\left\{-\delta^2 n\varepsilon/2\right\} \cdot nc + \left(1 - \exp\left\{-\delta^2 n\varepsilon/2\right\}\right) < 1 + \exp\left\{-\delta^2 n\varepsilon/2\right\} \cdot nc.$$

Using the inequality $xe^{-ax} \leq (ae)^{-1}$ for any $x, a > 0$, we have

$$\exp\left\{-\delta^2 n\varepsilon/2\right\} \cdot nc = n \cdot \exp\left\{-\delta^2 n\varepsilon/4\right\} \cdot \exp\left\{-\delta^2 n\varepsilon/4\right\} \cdot c \leq \frac{4}{\delta^2 \varepsilon e} \cdot \exp\left\{-\delta^2 n\varepsilon/4\right\} \cdot c.$$

Therefore,

$$\psi_1 < 1 + \frac{4}{\delta^2 \varepsilon e} \cdot \exp\left\{-\delta^2 n\varepsilon/4\right\} \cdot c \leq 1 + \frac{4c}{\delta^2 \varepsilon e} \cdot \exp\left\{-\delta^2 \varepsilon^2/(4c)\right\} \text{ since } nc \geq \varepsilon.$$

**Case 4:** When $\varepsilon \leq nc \leq (1 + \delta)^{-1}$, since $\mu = \mathbb{E}[\mathrm{Binom}(n, \varepsilon)] = n \cdot \varepsilon \leq \varepsilon \cdot (c(1 + \delta))^{-1}$, by Theorem A.2, we have

$$\mathbb{P}\left[\mathrm{Binom}(n, \varepsilon) \geq \varepsilon \cdot c^{-1}\right] \leq \mathbb{P}\left[\mathrm{Binom}(n, \varepsilon) \geq (1 + \delta) \cdot \mu\right] < \exp\left\{-\delta^2\mu/(2 + \delta)\right\} = \exp\left\{-\delta^2 n\varepsilon/(2 + \delta)\right\}. \tag{5}$$

Since the optimal cost is $nc$, by (4), the competitive ratio in this case is

$$
\begin{aligned}
\psi_2 := \frac{\mathbb{E}[\text{ALG}]}{nc} &< \left(1 - \exp\left\{-\delta^2 n\varepsilon/(2+\delta)\right\}\right) + \exp\left\{-\delta^2 n\varepsilon/(2+\delta)\right\} \cdot (nc)^{-1} \\
&< 1 + \varepsilon^{-1} \cdot \exp\left\{-\delta^2\varepsilon^2/(c(2+\delta))\right\} \text{ since } nc \geq \varepsilon.
\end{aligned}
\tag{6}
$$

Let $\psi$ denote the competitive ratio of the algorithm. From the four cases above, $\psi$ is bounded by the maximum of $(1-\delta)^{-1}$, $1+\delta$, $\psi_1$, and $\psi_2$.

We now let $\delta = c^{\frac{1}{4}}$ (any exponent $< 1/2$ would work). Then by (5) and (6),

$$
\psi_1 < 1 + \frac{4c^{\frac{1}{2}}}{\varepsilon e} \cdot \exp\left\{-\varepsilon^2/(4c^{\frac{1}{2}})\right\}, \psi_2 < 1 + \varepsilon^{-1} \cdot \exp\left\{\varepsilon^2/(c^{\frac{1}{2}}(2+\delta))\right\}.
$$

In particular, note that as $c \to 0$, we have $\psi \to 1$.

## C. Proof of Lemma 5.3

Since our result targets small $c$, assume $2c < \varepsilon$ and take $2c \leq \delta \leq \varepsilon$ in the following proof.

We introduce two definitions that will be useful in the analysis.

**Atoms in an instance.**  Recall that all trips $(c_i, t_i)$ satisfy $0 < c_i \leq c$. However, for two trips $(c_i, t_i)$ and $(c_j, t_j)$, the ratio $c_i/c_j$ can be arbitrarily large, which complicates the analysis. To address this, we partition the trips of an instance into groups, called *atoms*, each having total full cost between $c$ and $2c$, and perform our analysis at the level of atoms rather than individual trips.

Given a Bahncard instance $\mathcal{I}$, we construct a partition of its trips into disjoint spans, called atoms, as follows. Starting from the first trip $(c_1, t_1)$, let $i$ be the smallest index such that $\sum_{j=1}^{i} c_j \geq c$, and define $R_1 = \{(c_1, t_1), \ldots, (c_i, t_i)\}$. We then repeat this procedure starting from index $i+1$, continuing until the end of the instance. This yields a partition $\mathcal{R} = R_1, \ldots, R_\ell$, where each $R_k$ is called an *atom*. Since each $c_i \leq c$, it is immediate that $c \leq c_{\mathcal{I}}(R_k) \leq 2c$ for every atom except possibly the last one.

We say that a span $P$ is *atomic* if $P = R_i \cup \cdots \cup R_j$ for some $i \leq j$. For an atomic span, we treat its atoms as the basic units of trips.

For any span $P$ that starts in atom $R_i$ and ends in atom $R_j$, we define its *canonical atomic span* to be $P' = R_i \cup \cdots \cup R_j$, that is, the smallest atomic span containing all trips in $P$. Similarly, for a collection of spans $\mathcal{C}$, let $\mathcal{C}'$ denote the collection of canonical atomic spans corresponding to the spans in $\mathcal{C}$.

**Bad Spans.**  By construction, $\mathcal{S}$ has the property that every gap has length at least $\delta$ (except possibly at the beginning and the end of the instance). This motivates the following definition.

**Definition C.1.** We say that a span $P$ in instance $\mathcal{I}$ is of type 1, 2, or 3 if $\ell(P) \geq \delta$ and it satisfies, respectively,

1. $c(P) \geq 4(1+\delta)(1-\beta)^{-1}$;
2. $\delta \leq c(P) < 4(1+\delta)(1-\beta)^{-1}$;
3. $c(P) < \delta$.

Fix a sampled instance $\hat{\mathcal{I}}$. We call a span $P$ *bad* if $\ell(P) \geq \delta$ and one of the following holds:

1. $c(P) \geq 4(1+\delta)(1-\beta)^{-1}$ and $\hat{c}(P) < 4(1-\beta)^{-1}$;
2. $\delta \leq c(P) < 4(1+\delta)(1-\beta)^{-1}$ and $\hat{c}(P) < c(P) \cdot (1+\delta)^{-1}$;
3. $c(P) < \delta$.

We say that such bad spans are of type 1, 2, and 3, respectively. We call a span $P$ good if $\ell(P) \geq \delta$ and $P$ is not bad. We call an interval $[t, t']$ bad (resp. good) if the corresponding span $\mathcal{I}([t, t'])$ is bad (resp. good). For the solution $\mathcal{S}$, we denote by $\mathsf{bad}_1$, $\mathsf{bad}_2$, and $\mathsf{bad}_3$ the total rental cost incurred in bad gaps of types 1, 2, and 3, respectively. That is, for $i \in \{1, 2, 3\}$,

$$\mathsf{bad}_i = \sum_{\text{type } i \text{ bad gap } g} c(g).$$

Recall that we partition the trips into atoms so that the cost of each atom lies between $c$ and $2c$. Our goal is to analyze atomic spans rather than arbitrary spans. Accordingly, given a bad span $P$, we consider its canonical atomic span $P'$—the smallest atomic span containing all trips in $P$. We next characterize the possible forms of such atomic spans, which motivates the following definition.

**Definition C.2.** We say that an atomic span $P'$ in instance $\mathcal{I}$ is of *atomic type 1* or *atomic type 2* if $\ell(P') \geq \delta$ and it satisfies, respectively,

1. $c(P') \geq 4(1 + \delta)(1 - \beta)^{-1}$;

2. $\delta \leq c(P') < 4(1 + \delta)(1 - \beta)^{-1} + 4c$.

Fix a sample $\hat{\mathcal{I}}$. We call an atomic span $P'$ *bad of atomic type 1* or *bad of atomic type 2* if $\ell(P') \geq \delta$ and it satisfies, respectively,

1. $c(P') \geq 4(1 + \delta)(1 - \beta)^{-1}$ and $\hat{c}(P') < 4(1 - \beta)^{-1} + 4c$;

2. $\delta \leq c(P') < 4(1 + \delta)(1 - \beta)^{-1} + 4c$ and $\hat{c}(P') < c(P) \cdot (1 + \delta)^{-1} + 4c$.

It follows directly from the definitions that if $P$ is (bad) of type 1 or 2, then its canonical atomic span $P'$ is (bad) of atomic type 1 or 2. Note that it is possible for a bad span to be of both atomic type 1 and 2.

**Preprocessing of long gaps.** Fix a sample $\hat{\mathcal{I}}$. For the purpose of analysis, we split some long gaps and treat them as the union of several gaps.

Consider any long gap $g = (t, t')$ such that $\hat{c}((t, t')) > 4(1 - \beta)^{-1}$. We split $g$ as follows. In the $i$th iteration, if $\hat{c}((t, t')) < 4(1 - \beta)^{-1}$, we set $g_i \leftarrow (t, t')$ and halt. Otherwise, choose $t'' \in (t, t')$ minimum such that $\hat{c}([t, t'']) \geq (1 - \beta)^{-1}$. Set $g_i \leftarrow (t, t'']$ and update $t \leftarrow t''$ to continue.

By optimality of $\hat{\mathcal{S}}$ on $\hat{\mathcal{I}}$, $\hat{c}(t'') < (1 - \beta)^{-1}$ for each $t''$ in a long gap $(t, t')$. Therefore, $\hat{c}(g_i) < 2(1 - \beta)^{-1}$ for all $i$ except the last one. Therefore, the procedure above partitions $g$ sequentially into a new set of long gaps $g_1, \ldots, g_m$, so that $(1 - \beta)^{-1} \leq \hat{c}(g_i) < 2(1 - \beta)^{-1}$ for all $1 \leq i \leq m - 1$, and $(1 - \beta)^{-1} \leq \hat{c}(g_m) < 4(1 - \beta)^{-1}$. By optimality of $\hat{\mathcal{S}}$ on $\hat{\mathcal{I}}$, $\ell(g_i) \geq 1 > \delta$ for all $1 \leq i \leq m$. After applying this splitting procedure to every original long gap, all resulting long gaps satisfy $\ell(g) \geq \delta$ and $\hat{c}(g) < 4(1 - \beta)^{-1}$.

**Bounding the last factor.** We now compare $\mathrm{cost}(\mathcal{S}, \hat{\mathcal{I}})$ and $\mathrm{cost}(\mathcal{S}, \mathcal{I})$ for a fixed sample $\hat{\mathcal{I}}$. The buy costs are equal for the same solution, so it suffices to compare the rental cost. Moreover, in $\mathcal{S}$, all uncovered rental costs are in the long gaps.

Fix a long gap $g = (t, t')$ (or $g = (t, t']$, if $g$ is created from the splitting procedure above).

If $g$ is good, then $\hat{c}(g) < 4(1 - \beta)^{-1}$ by the splitting procedure. Hence $g$ must be of type 2, and we have $c(g) \leq (1 + \delta) \cdot \hat{c}(g)$.

If $g$ is bad of type 3, $c(g) < \delta$. Recall that if a gap $g'$ is produced by the splitting procedure, then $\hat{c}(g') \geq (1 - \beta)^{-1}$, so $c(g') \geq \varepsilon \cdot (1 - \beta)^{-1}$. Since we assumed that $\delta < \varepsilon$, such a gap $g'$ cannot be bad of type 3. Therefore, every gap of type 3 is not produced by splitting, which implies that there can be at most one such gap between every two consecutive cards. It follows that we may upper bound $\mathsf{bad}_3$ by $\delta \cdot \mathsf{buy}(\mathcal{S})$.

Consider the cost in the two gaps $[0, t_1)$ and $(t_k + 1, T]$, where $t_1$ is the first card and $t_k$ is the last card. Since their analysis are completely symmetric, we focus on $[0, t_1)$. If $\hat{c}([0, t_1)) \geq c([0, t_1)) \cdot (1 + \delta)^{-1}$, then the same bound for good gaps still applies. Otherwise, the cost is bounded by the following two terms. Define

$$\mathsf{bad}_- = \mathbb{1}[\hat{c}([0, t_1)) < \min\{(1 - \beta)^{-1}, c([0, t_1)) \cdot (1 + \delta)^{-1}\}] \cdot c([0, t_1))$$

$$\mathsf{bad}_+ = 1[\hat{c}((t_k+1,T)) < \min\{(1-\beta)^{-1}, c((t_k+1,T)) \cdot (1+\delta)^{-1}\}] \cdot c((t_k+1,T)).$$

By the bounds stated above,

$$\mathsf{rent}(\mathcal{S},\mathcal{I}) = (1-\beta) \cdot \sum_{\text{long gap } g} c(g) + \beta \cdot c(\mathcal{I})$$

$$\leq (1-\beta) \cdot \sum_{\text{good gap } g} c(g) + \beta \cdot c(\mathcal{I}) + \sum_{\text{bad gap } g} c(g) + \mathsf{bad}_- + \mathsf{bad}_+$$

$$\leq (1-\beta) \cdot (1+\delta) \cdot \sum_{\text{good gap } g} \hat{c}(g) + \beta \cdot c(\mathcal{I}) + \mathsf{bad}_3 + \mathsf{bad}_1 + \mathsf{bad}_2 + \mathsf{bad}_- + \mathsf{bad}_+$$

Note that $\beta \cdot c(\mathcal{I}) = \mathbb{E}[\beta \cdot \hat{c}(\mathcal{I})]$. Therefore,

$$\mathbb{E}[\mathsf{rent}(\mathcal{S},\mathcal{I})] \leq (1+\delta) \cdot \mathbb{E}\left[ (1-\beta) \cdot \sum_{\text{good gap } g} \hat{c}(g) + \beta \cdot \hat{c}(\mathcal{I}) \right] + \mathbb{E}[\mathsf{bad}_3 + \mathsf{bad}_1 + \mathsf{bad}_2 + \mathsf{bad}_- + \mathsf{bad}_+]$$

$$\leq (1+\delta) \cdot \mathbb{E}[\mathsf{rent}(\mathcal{S},\hat{\mathcal{I}})] + \mathbb{E}[\mathsf{bad}_3 + \mathsf{bad}_1 + \mathsf{bad}_2 + \mathsf{bad}_- + \mathsf{bad}_+].$$

Adding the buy cost and using the bound on $\mathsf{bad}_3$,

$$\mathbb{E}[\mathsf{cost}(\mathcal{S},\mathcal{I})] = \mathbb{E}[\mathsf{buy}(\mathcal{S})] + \mathbb{E}[\mathsf{rent}(\mathcal{S},\mathcal{I})]$$

$$\leq \mathbb{E}[\mathsf{buy}(\mathcal{S})] + (1+\delta) \cdot \mathbb{E}[\mathsf{rent}(\mathcal{S},\hat{\mathcal{I}})] + \mathbb{E}[\mathsf{bad}_3 + \mathsf{bad}_1 + \mathsf{bad}_2 + \mathsf{bad}_- + \mathsf{bad}_+]$$

$$\leq \mathbb{E}[\mathsf{buy}(\mathcal{S})] + (1+\delta) \cdot \mathbb{E}[\mathsf{rent}(\mathcal{S},\hat{\mathcal{I}})] + \delta \cdot \mathbb{E}[\mathsf{buy}(\mathcal{S})] + \mathbb{E}[\mathsf{bad}_1 + \mathsf{bad}_2 + \mathsf{bad}_- + \mathsf{bad}_+]$$

$$= (1+\delta) \cdot \mathbb{E}[\mathsf{cost}(\mathcal{S},\hat{\mathcal{I}})] + \mathbb{E}[\mathsf{bad}_1 + \mathsf{bad}_2 + \mathsf{bad}_- + \mathsf{bad}_+]$$

Separating the expectation of each term, we now have

$$\mathbb{E}[\mathsf{cost}(\mathcal{S},\mathcal{I})] \leq (1+\delta) \cdot \mathbb{E}[\mathsf{cost}(\mathcal{S},\hat{\mathcal{I}})] + \mathbb{E}[\mathsf{bad}_1] + \mathbb{E}[\mathsf{bad}_2] + \mathbb{E}[\mathsf{bad}_-] + \mathbb{E}[\mathsf{bad}_+]. \tag{7}$$

**Bounds for bad cost.** It remains to bound $\mathbb{E}[\mathsf{bad}_1]$, $\mathbb{E}[\mathsf{bad}_2]$, $\mathbb{E}[\mathsf{bad}_-]$, and $\mathbb{E}[\mathsf{bad}_+]$. We will first bound $\mathbb{E}[\mathsf{bad}_1]$ and $\mathbb{E}[\mathsf{bad}_2]$ with respect to $\mathbb{E}[\mathsf{cost}(\mathcal{S},\hat{\mathcal{I}})]$, and then bound $\mathbb{E}[\mathsf{bad}_-]$ and $\mathbb{E}[\mathsf{bad}_+]$ with similar techniques.

Directly bounding $\mathsf{bad}_1$ and $\mathsf{bad}_2$ is hard, since the randomness of the sampling affects the samples and $\mathcal{S}$ simultaneously. Instead, we bound relaxed quantities that upper bound $\mathsf{bad}_1$ and $\mathsf{bad}_2$ and are independent of the algorithm. We can then bound $\mathbb{E}[\mathsf{bad}_1]$ and $\mathbb{E}[\mathsf{bad}_2]$ with respect to $\mathbb{E}[\mathsf{cost}(\mathcal{S},\hat{\mathcal{I}})]$ by considering two partitions of the original instance $\mathcal{I}$ into atomic spans.

We partition the original instance $\mathcal{I}$ from the left into a collection of minimal atomic type 2 spans. Precisely, this means the following. Starting from atom $R_1$, if there exists an atomic type 2 span that starts at $R_1$, we mark the shortest such span $Q_1 = \{R_1, \ldots, R_j\}$ and proceed to $R_{j+1}$. If no such span exists, proceed to the next atom $R_2$. We repeat this until the end of instance is reached. This yields a collection of $k_2$ atomic spans $\mathcal{Q} = \{Q_1, \ldots, Q_{k_2}\}$.

**Lemma C.3.** *For the $k_2$ defined above,*

$$\mathbb{E}[\mathsf{bad}_2] \leq k_2 \cdot \frac{100(1+\delta)^3}{c^2(1-\beta)^3} \cdot \exp\left\{ -\frac{2\varepsilon^2\delta^3}{c(1+2\delta)^2} \right\}.$$

*Proof.* Fix a sample. Define $\mathsf{bad}_2'$ to be the maximum total cost of a disjoint collection of bad spans of type 2 in the original instance. In other word, for $\mathcal{P}$ taken over disjoint collections of type 2 bad spans,

$$\mathsf{bad}_2' = \max_{\mathcal{P}} \sum_{P \in \mathcal{P}} c(P).$$

Note that unlike $\mathsf{bad}_2$, $\mathsf{bad}_2'$ doesn't have dependence on $\mathcal{S}$. Clearly, we have $\mathsf{bad}_2 \leq \mathsf{bad}_2'$, since the collection of type 2 bad gaps is disjoint. It then suffices to bound $\mathbb{E}[\mathsf{bad}_2']$.

For $\mathcal{P}$ a disjoint collection of type 2 bad span, consider $\mathcal{P}'$, the canonical collection of atomic spans corresponding to $\mathcal{P}$. For $i \neq j$, $\mathcal{P}$ contains at most one span that starts and ends in $R_i$ and $R_j$. When $i = j$, since $c(R_i) \leq 2c$ for all $i$, no type 2 bad span is possible. As a result, although spans in $\mathcal{P}'$ may not be disjoint, these spans are distinct.

For any $\mathcal{P}$, each span $P \in \mathcal{P}$ has $c(P) \leq 4(1 + \delta)(1 - \beta)^{-1}$. Since $|\mathcal{P}| = |\mathcal{P}'|$, we have

$$\mathsf{bad}_2' \leq \frac{4(1 + \delta)}{1 - \beta} \cdot [\text{\# of atomic type 2 bad } P'].$$

Taking expectation with respect to sampling, we have

$$\mathbb{E}[\mathsf{bad}_2'] \leq \frac{4(1 + \delta)}{1 - \beta} \cdot \sum_{\text{atomic type 2 } P'} \mathbb{P}[P' \text{ is bad}]$$

By the expression above, it remains to bound the total number of atomic type 2 spans and the probability for any atomic type 2 span to be bad. We first bound the probability using Hoeffding's inequality from Theorem A.1. Let $P'$ be bad of atomic type 2. By definition,

$$\mathbb{P}[P' \text{ is bad}] = \mathbb{P}\left[\hat{c}(P') < \frac{c(P')}{1 + \delta} + 4c\right] = \mathbb{P}\left[|\hat{c}(P') - c(P')| > \frac{\delta}{1 + \delta} \cdot c(P') - 4c\right].$$

For sufficiently small $c$,

$$\mathbb{P}\left[|\hat{c}(P') - c(P')| > \frac{\delta}{1 + \delta} \cdot c(P') - 4c\right] \leq \mathbb{P}\left[|\hat{c}(P') - c(P')| > \frac{\delta}{1 + 2\delta} \cdot c(P')\right].$$

By Hoeffding's inequality,

$$\mathbb{P}\left[|\hat{c}(P') - c(P')| > \frac{\delta}{1 + 2\delta} \cdot c(P')\right] < 2 \exp\left\{-\frac{2\varepsilon^2 \cdot \left(\frac{\delta}{1 + 2\delta} \cdot c(P')\right)^2}{\sum_{(c_i, t_i) \in P'} c_i^2}\right\}.$$

Recall that each $c_i < c$ and $\sum_{(c_i, t_i) \in P'} c_i = c(P')$. Therefore,

$$\mathbb{P}\left[|\hat{c}(P') - c(P')| > \frac{\delta}{1 + 2\delta} \cdot c(P')\right] < 2 \exp\left\{-\frac{2\varepsilon^2 \cdot \left(\frac{\delta}{1 + 2\delta} \cdot c(P')\right)^2}{c \cdot c(P')}\right\} = 2 \exp\left\{-\frac{2\varepsilon^2 \cdot \left(\frac{\delta}{1 + 2\delta}\right)^2 \cdot c(P')}{c}\right\}.$$

Since $P'$ is of atomic type 2, $c(P') \geq \delta$. Therefore,

$$\mathbb{P}\left[|\hat{c}(P') - c(P')| > \frac{\delta}{1 + 2\delta} \cdot c(P')\right] < 2 \exp\left\{-\frac{2\varepsilon^2 \cdot \left(\frac{\delta}{1 + 2\delta}\right)^2 \cdot \delta}{c}\right\}$$

We now bound the number of spans $P'$ of atomic type 2 satisfying the first two conditions.

Since each marked span $Q_i$ is of atomic type 2, $c(Q_i) \leq 4(1 + \delta)(1 - \beta)^{-1} + 4c \leq 5(1 + \delta)(1 - \beta)^{-1}$. This implies that each $Q_i$ contains at most $5(1 + \delta)((1 - \beta)c)^{-1}$ atoms. Therefore, at most $k_2 \cdot 5(1 + \delta)((1 - \beta)c)^{-1}$ atoms in $\mathcal{I}$ are covered by some marked span $Q_i$.

Note that no span $P'$ of atomic type 2 can start at an atom $R_i$ not covered by a marked span. If $R_i$ is uncovered, the marking procedure reaches $R_i$ at some point, and because it is not marked, it doesn't have a type 2 span starting from it.

Furthermore, fixing a starting atom $R_i$, there are at most $4(1 + \delta)((1 - \beta)c)^{-1} + 4 \leq 5(1 + \delta)((1 - \beta)c)^{-1}$ choices of an ending atom $R_j$ such that $\{R_i, \ldots, R_j\}$ is a span of type 2 due to the bound on its total full cost. Therefore,

$$[\#P' \text{ of atomic type 2}] \leq \sum_{R_i \text{ covered by marked spans}} [\text{\# choices of ending trips}]$$

$$\leq k_2 \cdot \frac{5(1 + \delta)}{c(1 - \beta)} \cdot \frac{5(1 + \delta)}{c(1 - \beta)} = k_2 \cdot \frac{25(1 + \delta)^2}{c^2(1 - \beta)^2}.$$

This gives the bound

$$\mathbb{E}[\mathsf{bad}'_2] \leq k_2 \cdot \frac{4(1+\delta)}{1-\beta} \cdot \frac{25(1+\delta)^2}{c^2(1-\beta)^2} \cdot \exp\left\{-\frac{2\varepsilon^2\delta^3}{c(1+2\delta)^2}\right\}. \tag{8}$$

Lemma C.3 then follows from $\mathsf{bad}_2 \leq \mathsf{bad}'_2$. $\qquad\square$

Similar to the discussion on $\mathsf{bad}_2$, partition the original instance from the left into a collection of minimal atomic type 1 spans. Precisely, this means the following. Starting from atom $R_1$, if there exists an atomic type 1 span that starts at $R_1$, we mark the shortest such span $Q_1 = \{R_1, \ldots, R_j\}$ and proceed to $R_{j+1}$, so that $\{R_1, \ldots, R_{j+1}\}$ is not of type 1. If no such span exists, proceed to the next atom $R_2$. We repeat this until the end of instance is reached. This gives a collection of spans $\mathcal{Q} = \{Q_1, \ldots, Q_{k_1}\}$.

**Lemma C.4.** *For the $k_1$ defined above,*

$$\mathbb{E}[\mathsf{bad}_1] \leq k_1 \cdot \frac{c(1+2\delta)^2}{2e\varepsilon^2\delta^2} \cdot \exp\left\{-\frac{8\varepsilon^2\delta^2(1+\delta)}{c(1+2\delta)^2(1-\beta)}\right\} \cdot \left(1 - \exp\left\{\frac{2\varepsilon^2\delta^2}{(1+2\delta)^2}\right\}\right)^{-1}$$
$$\cdot \left(2 + \left(2 + \frac{4(1+\delta)}{(1-\beta)c}\right) \cdot \left(1 - \exp\left\{-\frac{8\varepsilon^2\delta^2(1+\delta)}{c(1+2\delta)^2(1-\beta)}\right\}\right)^{-1}\right)$$

*Proof.* Fix a sample. Define $\mathsf{bad}'_1$ to be the maximum total cost of a disjoint collection of type 1 bad spans, namely for $\mathcal{P}$ taken over disjoint collections of type 1 bad spans,

$$\mathsf{bad}'_1 = \max_{\mathcal{P}} \sum_{P \in \mathcal{P}} c(P).$$

Unlike $\mathsf{bad}_1$, $\mathsf{bad}'_1$ doesn't have dependence on $\mathcal{S}$. Clearly, we have $\mathsf{bad}_1 \leq \mathsf{bad}'_1$ since the collection of type 1 bad gaps is disjoint. It then suffices to bound $\mathbb{E}[\mathsf{bad}'_1]$.

By the same argument from the proof of Lemma C.3, spans in $\mathcal{P}'$ must be distinct.

Since $c(\mathcal{P}') \geq c(\mathcal{P})$, it suffices to bound the total cost of type 1 atomic spans. By the rules of constructing $\mathcal{Q}$, every atomic type 1 span must start in some $Q_i$. Therefore,

$$\mathbb{E}[\mathsf{bad}'_1] \leq \sum_{i=1}^{k_1} \sum_{\text{atomic type 1 span } P' \text{ that starts in } Q_i} \mathbb{P}\left[P' \text{ is bad of atomic type 1}\right] \cdot c(P').$$

We first bound $\mathbb{P}\left[P' \text{ is bad of atomic type 1}\right] \cdot c(P')$. By the same Chernoff bound used for $\mathsf{bad}'_2$, we have

$$\mathbb{P}\left[P' \text{ is bad of atomic type 1}\right] < \exp\left\{-\frac{2\varepsilon^2\delta^2 c(P')}{c(1+2\delta)^2}\right\}.$$

By the inequality $xe^{-ax} \leq (ae)^{-1}$ for $x, a > 0$, we have

$$\mathbb{P}\left[P' \text{ is bad of atomic type 1}\right] \cdot c(P') < \frac{c(1+2\delta)^2}{2e\varepsilon^2\delta^2} \cdot \exp\left\{-\frac{\varepsilon^2\delta^2 c(P')}{c(1+2\delta)^2}\right\}. \tag{9}$$

We now bound $\sum_{\text{atomic type 1 span } P' \text{ that starts in } Q_i} \mathbb{P}\left[P' \text{ is bad of atomic type 1}\right] \cdot c(P')$. Partition $Q_i$ into smaller atomic spans as follows. Starting from $R_1$. Find $u$ minimal such that $c(R_1 \cup \cdots \cup R_u) \geq 4(1+\delta)(1-\beta)^{-1}$ and proceed to $R_{u+1}$. Repeat this procedure until $Q_i$ is partitioned into $\{Q_{i,1}, \ldots, Q_{i,p}\}$. By construction, for $j < p$, $4(1+\delta)(1-\beta)^{-1} \leq c(Q_{i,j}) \leq 4(1+\delta)(1-\beta)^{-1} + 2c$.

Fix some atom $R_u$ in $Q_{i,j}$ where $j \leq p - 2$. By minimality of $Q_i$, any atomic type 1 span $P'$ that starts at $R_u$ contains all trips in $Q_{i,j+1}, \ldots, Q_{i,p}$. So, for such span $P'$, $c(P') \geq (p - j - 1) \cdot 4 \cdot (1+\delta) \cdot (1-\beta)^{-1}$. Let $R_u \cup \cdots \cup R_v$ be the minimal such span. Then, we have

$$\{\text{type 1 spans that start at } R_u\} \subset \{R_u \cup \cdots \cup R_{v'} : v' \geq v\}.$$

Therefore, by (9),

$$\sum_{\text{atomic type 1 span } P' \text{ that starts at } R_u} \mathbb{P}\left[P' \text{ is bad of atomic type } 1\right] \cdot c(P')$$

$$\leq \sum_{v' \geq v} \mathbb{P}\left[R_u \cup \cdots \cup R_v \text{ is bad of atomic type } 1\right] \cdot c(R_u \cup \cdots \cup R_v)$$

$$< \frac{c(1+2\delta)^2}{2e\varepsilon^2\delta^2} \cdot \sum_{i=0}^{\infty} \exp\left\{-\frac{2\varepsilon^2\delta^2}{c(1+2\delta)^2} \cdot \left(\frac{(p-j-1)\cdot 4(1+\delta)}{1-\beta} + ci\right)\right\}$$

$$= \frac{c(1+2\delta)^2}{2e\varepsilon^2\delta^2} \cdot \exp\left\{-\frac{8\varepsilon^2\delta^2(1+\delta)(p-j-1)}{c(1+2\delta)^2(1-\beta)}\right\} \cdot \left(1 - \exp\left\{-\frac{2\varepsilon^2\delta^2}{(1+2\delta)^2}\right\}\right)^{-1}$$

Counting all trips in $Q_{i,j}$, we have

$$\sum_{\text{atomic type 1 span } P' \text{ that starts in } Q_{i,j}} \mathbb{P}\left[P' \text{ is bad of atomic type } 1\right] \cdot c(P)$$

$$\leq \frac{c(1+2\delta)^2}{2e\varepsilon^2\delta^2} \cdot \left(2 + \frac{4(1+\delta)}{(1-\beta)c}\right) \cdot \exp\left\{-\frac{8\varepsilon^2\delta^2(1+\delta)(p-j-1)}{c(1+2\delta)^2(1-\beta)}\right\} \cdot \left(1 - \exp\left\{-\frac{2\varepsilon^2\delta^2}{(1+2\delta)^2}\right\}\right)^{-1} \quad (10)$$

Separately consider the last two spans $Q_{i,p-1}$ and $Q_{i,p}$. Note that for an atomic type 1 span $P'$ to start in these two spans, it must hold that $c(P') \geq 4(1+\delta)(1-\beta)^{-1}$ by definition of atomic type 1 spans. Therefore, by the same bounds above, we have

$$\sum_{\text{type 1 span } I \text{ that starts in } P_{i,p-1} \text{ and } P_{i,p}} \mathbb{P}\left[P' \text{ is bad of atomic type } 1\right] \cdot c(P)$$

$$\leq \frac{c(1+2\delta)^2}{2e\varepsilon^2\delta^2} \cdot \exp\left\{-\frac{8\varepsilon^2\delta^2(1+\delta)}{c(1+2\delta)^2(1-\beta)}\right\} \cdot \left(1 - \exp\left\{-\frac{2\varepsilon^2\delta^2}{(1+2\delta)^2}\right\}\right)^{-1} \quad (11)$$

Now counting all trips in $Q_i$, we have

$$\sum_{\text{atomic type 1 span } P' \text{ that starts in } Q_i} \mathbb{P}\left[P' \text{ is bad of atomic type } 1\right] \cdot c(P')$$

$$= \sum_{j=1}^{p-2} \sum_{\text{atomic type 1 span } P' \text{ that starts in } Q_{i,j}} \mathbb{P}\left[P' \text{ is bad of atomic type } 1\right] \cdot c(P')$$

$$+ \sum_{\text{atomic type 1 span } P' \text{ that starts in } Q_{i,p-1} \text{ and } Q_{i,p}} \mathbb{P}\left[P' \text{ is bad of atomic type } 1\right] \cdot c(P')$$

$$\leq \frac{c(1+2\delta)^2}{2e\varepsilon^2\delta^2} \cdot \exp\left\{-\frac{8\varepsilon^2\delta^2(1+\delta)}{c(1+2\delta)^2(1-\beta)}\right\} \cdot \left(1 - \exp\left\{-\frac{2\varepsilon^2\delta^2}{(1+2\delta)^2}\right\}\right)^{-1}$$

$$\cdot \left(2 + \left(2 + \frac{4(1+\delta)}{(1-\beta)c}\right) \cdot \left(1 - \exp\left\{-\frac{8\varepsilon^2\delta^2(1+\delta)}{c(1+2\delta)^2(1-\beta)}\right\}\right)^{-1}\right)$$

There are a total of $k_1$ of $Q_i$, so we have

$$\mathbb{E}[\mathsf{bad}_1'] \leq k_1 \cdot \frac{c(1+2\delta)^2}{2e\varepsilon^2\delta^2} \cdot \exp\left\{-\frac{8\varepsilon^2\delta^2(1+\delta)}{c(1+2\delta)^2(1-\beta)}\right\} \cdot \left(1 - \exp\left\{-\frac{2\varepsilon^2\delta^2}{(1+2\delta)^2}\right\}\right)^{-1}$$

$$\cdot \left(2 + \left(2 + \frac{4(1+\delta)}{(1-\beta)c}\right) \cdot \left(1 - \exp\left\{-\frac{8\varepsilon^2\delta^2(1+\delta)}{c(1+2\delta)^2(1-\beta)}\right\}\right)^{-1}\right) \quad (12)$$

Lemma C.4 then follows from $\mathsf{bad}_1 \leq \mathsf{bad}_1'$. □

**Lemma C.5.**

$$\mathbb{E}[\mathsf{bad}_-] + \mathbb{E}[\mathsf{bad}_+] \leq 2\delta + \frac{2(1+\delta)^2}{(1-\beta)^2 c} \cdot \exp\left\{-\frac{2\varepsilon^2\delta^3}{c(1+2\delta)^2}\right\}$$

$$+ 2\exp\left\{-\frac{8\varepsilon^2\delta^2(1+\delta)}{c(1+2\delta)^2(1-\beta)}\right\} \cdot \left(1 - \exp\left\{-\frac{2\varepsilon^2\delta^2}{(1+2\delta)^2}\right\}\right)^{-1}. \qquad (13)$$

*Proof.* It suffices to bound $\mathbb{E}[\mathsf{bad}_-]$, since the exact same bound applies to $\mathbb{E}[\mathsf{bad}_+]$ by symmetry. Define $\mathsf{bad}'_-$ by

$$\mathsf{bad}'_- = \sup_{t\in[0,T]} 1\left[\hat{c}([0,t)) < \min\left\{\frac{1}{1-\beta}, \frac{c([0,t))}{1+\delta}\right\}\right] \cdot c([0,t)),$$

which upper bounds $\mathsf{bad}_-$. Let $(c,t)$ be the first trip in the original instance $\mathcal{I}$ and $P$ ranging over spans containing $(c,t)$. Then we have

$$\mathbb{E}[\mathsf{bad}'_-] \leq \mathbb{E}\left[\max_{P\ni(c,t)}\left\{1\left[\hat{c}(P) < \min\left\{\frac{1}{1-\beta}, \frac{c(P)}{1+\delta}\right\}\right] \cdot c(P)\right\}\right]$$

$$\leq \delta + \mathbb{E}\left[\max_{P\ni(c,t):c(P)\geq\delta}\left\{1\left[\hat{c}(P) < \min\left\{\frac{1}{1-\beta}, \frac{c(P)}{1+\delta}\right\}\right] \cdot c(P)\right\}\right]$$

Consider canonical atomic span $P'$ corresponding to each $P$. Then, we have the bound

$$\max_{P\ni(c,t):c(P)\geq\delta}\left\{1\left[\hat{c}(P) < \min\left\{\frac{1}{1-\beta}, \frac{c(P)}{1+\delta}\right\}\right] \cdot c(P)\right\} \leq \max_{P\ni(c,t):c(P)\geq\delta}\left\{1\left[\hat{c}(P') < \min\left\{\frac{1}{1-\beta}+c, \frac{c(P')}{1+\delta}+c\right\}\right] \cdot c(P')\right\}$$

Therefore, we have the bound

$$\mathbb{E}[\mathsf{bad}'_-] \leq \delta + \mathbb{E}\left[\max_{P\ni(c,t):c(P)\geq\delta}\left\{1\left[\hat{c}(P') < \min\left\{\frac{1}{1-\beta}+c, \frac{c(P')}{1+\delta}+c\right\}\right] \cdot c(P')\right\}\right]$$

$$\leq \delta + \sum_{P'\ni(c,t):c(P)\geq\delta}\mathbb{E}\left[1\left[\hat{c}(P') < \min\left\{\frac{1}{1-\beta}+c, \frac{c(P')}{1+\delta}+c\right\}\right] \cdot c(P')\right]$$

$$= \delta + \sum_{P'\ni(c,t):c(P)\geq\delta}\mathbb{P}\left[\hat{c}(P') < \min\left\{\frac{1}{1-\beta}+c, \frac{c(P')}{1+\delta}+c\right\}\right] \cdot c(P')$$

$$= \delta + \sum_{P'\ni(c,t):\delta\leq c(P)<(1+\delta)/(1-\beta)}\mathbb{P}\left[\hat{c}(P') < \frac{c(P')}{1+\delta}+c\right] \cdot c(P')$$

$$+ \sum_{P'\ni(c,t):c(P)\geq(1+\delta)/(1-\beta)}\mathbb{P}\left[\hat{c}(P') < \frac{1}{1-\beta}+c\right] \cdot c(P')$$

Consider $P'$ for which $\delta \leq c(P') < (1+\delta)(1-\beta)^{-1}$. There are at most $(1+\delta)(c(1-\beta))^{-1}$ such atomic spans $P'$, and the probability can be bounded by Hoeffding's bound in the same way as the proof of Lemma C.3:

$$\mathbb{P}\left[\hat{c}(P') < \frac{c(P')}{1+\delta}+c\right] < \exp\left\{-\frac{2\varepsilon^2\delta^3}{c(1+2\delta)^2}\right\}.$$

For $P'$ such that $c(P') \geq (1+\delta)(1-\beta)^{-1}$, equation 11 gives the required bound since all definitions coincide:

$$\sum_{P'\ni(c,t):c(P)\geq(1+\delta)/(1-\beta)}\mathbb{P}\left[\hat{c}(P') < \frac{1}{1-\beta}+c\right] \cdot c(P') < \exp\left\{-\frac{8\varepsilon^2\delta^2(1+\delta)}{c(1+2\delta)^2(1-\beta)}\right\} \cdot \left(1 - \exp\left\{-\frac{2\varepsilon^2\delta^2}{(1+2\delta)^2}\right\}\right)^{-1}.$$

Combining the bounds above, we have

$$
\mathbb{E}[\mathsf{bad}'_-] \leq \delta + \frac{(1+\delta)^2}{(1-\beta)^2 c} \cdot \exp\left\{-\frac{2\varepsilon^2\delta^3}{c(1+2\delta)^2}\right\}
$$
$$
+ \exp\left\{-\frac{8\varepsilon^2\delta^2(1+\delta)}{c(1+2\delta)^2(1-\beta)}\right\} \cdot \left(1 - \exp\left\{-\frac{2\varepsilon^2\delta^2}{(1+2\delta)^2}\right\}\right)^{-1} \tag{14}
$$

Lemma C.5 then follows from $\mathsf{bad}_- \leq \mathsf{bad}'_-$ and $\mathsf{bad}_+$ having an identical bound. $\qquad \square$

**Lemma C.6.** *Both $k_1/\mathbb{E}[\mathsf{cost}(\mathcal{S}, \hat{\mathcal{I}})]$ and $k_2/\mathbb{E}[\mathsf{cost}(\mathcal{S}, \hat{\mathcal{I}})]$ are upper bounded by $(1+\delta)/\delta$.*

*Proof.* We now bound $\mathbb{E}[\mathsf{cost}(\mathcal{S}, \hat{\mathcal{I}})]$ with $k_1$ and $k_2$. We will only do this for $k_2$, and the same bound applies for $k_1$. Recall that the total cost of each atomic type 2 span is at least $\delta$. By the same bound from the proof of Lemma C.3, the total cost of sampled trips in each span is at least $\delta/(1+\delta)$ with probability $1 - o(1)$. In this case, if no card intersects the span, $\mathcal{S}$ pays $\delta/(1+\delta)$. Otherwise, the span accounts for $\delta/(1+\delta)$ from the card cost, since a card can overlap with at most $\lceil 1/\delta \rceil$ atomic type 2 spans. Therefore,

$$
\mathbb{E}[\mathsf{cost}(\mathcal{S}, \hat{\mathcal{I}})] \geq k_2 \cdot \frac{\delta}{1+\delta}.
$$

Therefore, both $k_1/\mathbb{E}[\mathsf{cost}(\mathcal{S}, \hat{\mathcal{I}})]$ and $k_2/\mathbb{E}[\mathsf{cost}(\mathcal{S}, \hat{\mathcal{I}})]$ are upper bounded by $(1+\delta)/\delta$. $\qquad \square$

We now choose proper $\delta$ for the algorithm.

**Lemma C.7.** *For $\delta = c^{\frac{1}{5}}$,*

$$
\frac{\mathbb{E}[\mathsf{bad}_2]}{\mathbb{E}[\mathsf{cost}(\mathcal{S}, \hat{\mathcal{I}})]} \leq \frac{100(1 + c^{\frac{1}{5}})^4}{c^{\frac{11}{5}}(1 - \beta)^3} \cdot \exp\left\{-\frac{2\varepsilon^2}{c^{\frac{2}{5}}(1 + 2c^{\frac{1}{5}})^3}\right\} =: \phi_2.
$$

*In particular, $\phi_2 \to 0$ as $c \to 0$.*

*Proof.* Immediately follows from Lemma C.3 and Lemma C.6. $\qquad \square$

**Lemma C.8.** *For $\delta = c^{\frac{1}{5}}$,*

$$
\frac{\mathbb{E}[\mathsf{bad}_1]}{\mathbb{E}[\mathsf{cost}(\mathcal{S}, \hat{\mathcal{I}})]} \leq \frac{c^{\frac{2}{5}}(1 + 2c^{\frac{1}{5}})^2(1 + c^{\frac{1}{5}})}{2e\varepsilon^2} \cdot \exp\left\{-\frac{8\varepsilon^2(1 + c^{\frac{1}{5}})}{c^{\frac{3}{5}}(1 + 2c^{\frac{1}{5}})^2(1 - \beta)}\right\} \cdot \left(1 - \exp\left\{-\frac{2\varepsilon^2 c^{\frac{2}{5}}}{(1 + 2c^{\frac{1}{5}})^2}\right\}\right)^{-1}
$$
$$
\cdot \left(2 + \left(2 + \frac{4(1 + c^{\frac{1}{5}})}{(1 - \beta)c}\right) \cdot \left(1 - \exp\left\{-\frac{8\varepsilon^2(1 + c^{\frac{1}{5}})}{c^{\frac{3}{5}} \cdot (1 + 2c^{\frac{1}{5}})^2 \cdot (1 - \beta)}\right\}\right)^{-1}\right) =: \phi_1.
$$

*In particular, $\phi_1 \to 0$ as $c \to 0$.*

*Proof.* Immediately follows from Lemma C.4 and Lemma C.6. $\qquad \square$

**Lemma C.9.** *For $\delta = c^{\frac{1}{5}}$,*

$$
\mathbb{E}[\mathsf{bad}_-] + \mathbb{E}[\mathsf{bad}_+] \leq 2c^{\frac{1}{5}} + \frac{2(1 + c^{\frac{1}{5}})^2}{(1 - \beta)^2 c} \cdot \exp\left\{-\frac{2\varepsilon^2}{c^{\frac{2}{5}}(1 + 2c^{\frac{1}{5}})^2}\right\}
$$
$$
+ 2\exp\left\{-\frac{8\varepsilon^2(1 + c^{\frac{1}{5}})}{c^{\frac{3}{5}}(1 + 2c^{\frac{1}{5}})^2(1 - \beta)}\right\} \cdot \left(1 - \exp\left\{-\frac{2\varepsilon^2 c^{\frac{2}{5}}}{(1 + 2c^{\frac{1}{5}})^2}\right\}\right)^{-1} =: \phi_3. \tag{15}
$$

*In particular, $\phi_3 \to 0$ as $c \to 0$.*

**Lemma C.10** (Lemma 5.3 restated). *Let $\phi = \phi_1 + \phi_2$. By setting $\delta = c^{\frac{1}{5}}$,*

$$\frac{\mathbb{E}[\mathsf{cost}(\mathcal{S}, \mathcal{I})]}{\mathbb{E}[\mathsf{cost}(\mathcal{S}, \hat{\mathcal{I}})]} \leq 1 + \phi + c^{\frac{1}{5}}.$$

*In particular, Algorithm 5.2 is $(3/2)$-competitive as $c \to 0$.*

*Proof.* Immediately follows from Lemma C.7, Lemma C.8, Lemma C.9, and Equation 7. □

## D. Proof of Theorem 5.4

Denote $\mathcal{S}^* = \mathrm{OPT}(\mathcal{I})$. To prove Theorem 5.4, we analyze $(\mathbb{E}[\mathsf{cost}(\mathcal{S}_{\mathrm{on}}, \hat{\mathcal{I}})] - b)/\mathsf{cost}(\mathcal{S}^*, \hat{\mathcal{I}})$.

For $\mathsf{buy}(\mathcal{S}_{\mathrm{on}})$, we count only the cost of cards that are already present in $\mathcal{S}_{\mathrm{off}}$; the cost of cards purchased online is accounted for in $\mathsf{rent}(\mathcal{S}_{\mathrm{on}})$. We denote by $\mathsf{short}(\mathcal{S}_{\mathrm{on}}, \mathcal{I})$ the rental cost paid by $\mathcal{S}_{\mathrm{on}}$ in short gaps of $\mathcal{S}_{\mathrm{off}}$ and cost of cards bought by $\mathcal{S}_{\mathrm{on}}$ online on instance $\mathcal{I}$. We further denote by $\mathsf{short}(\mathcal{S}_{\mathrm{off}}, \mathcal{I})$ the rental cost paid by $\mathcal{S}_{\mathrm{off}}$ in short gaps of $\mathcal{S}_{\mathrm{off}}$. If $\mathcal{S}_{\mathrm{off}}$ doesn't cover an interval between two long gaps, we say the interval is a skip interval.

We say a card at time $t$ is left-aligned if there is at least one trip at time $t$, and right-aligned if there is at least one trip at time $t + 1$. In an interval $[t, t']$ between two long gaps, we call the card that starts at $t$ the starting card, and the card that ends at $t'$ the ending card.

To bound the competitive ratio of Algorithm 5.3, we bound the following four factors:

$$\frac{\mathbb{E}[\mathsf{cost}(\mathcal{S}^*, \hat{\mathcal{I}})]}{\mathsf{cost}(\mathcal{S}^*, \mathcal{I})}, \frac{\mathsf{cost}(\hat{\mathcal{S}}, \hat{\mathcal{I}})}{\mathsf{cost}(\mathcal{S}^*, \hat{\mathcal{I}})}, \frac{\mathsf{cost}(\mathcal{S}_{\mathrm{off}}, \hat{\mathcal{I}})}{\mathsf{cost}(\hat{\mathcal{S}}, \hat{\mathcal{I}})}, \frac{\mathbb{E}[\mathsf{cost}(\mathcal{S}_{\mathrm{on}}, \mathcal{I})]}{\mathbb{E}[\mathsf{cost}(\mathcal{S}_{\mathrm{off}}, \hat{\mathcal{I}})]}.$$

**Lemma D.1.**

$$\mathbb{E}[\mathsf{cost}(\mathcal{S}_{\mathrm{off}}, \hat{\mathcal{I}})] \leq (1 + 2\delta) \cdot \mathsf{cost}(\mathcal{S}^*, \mathcal{I}).$$

*Proof.* The first two bounds from Lemma 5.2 still holds for the new algorithm.

By construction of $\mathcal{S}_{\mathrm{off}}$, all trips covered by $\hat{\mathcal{S}}$ are also covered by $\mathcal{S}_{\mathrm{off}}$, so $\mathsf{rent}(\mathcal{S}_{\mathrm{off}}, \hat{\mathcal{I}}) \leq \mathsf{rent}(\hat{\mathcal{S}}, \hat{\mathcal{I}})$. For the time interval between every two long gaps, if $\hat{\mathcal{S}}$ buys $k$ cards for $k \geq 1/\delta$, $\mathcal{S}_{\mathrm{off}}$ buys at most $k + \lceil ((k-1)\delta \rceil$ cards. By the lower bound on $k$, we have

$$\frac{k + \lceil (k-1)\delta \rceil}{k} \leq \frac{k + (k-1)\delta + 1}{k} \leq 1 + 2\delta.$$

So, $\mathsf{buy}(\mathcal{S}_{\mathrm{off}}) \leq (1 + 2\delta) \cdot \mathsf{buy}(\hat{\mathcal{S}})$. This implies $\mathsf{cost}(\mathcal{S}_{\mathrm{off}}, \hat{\mathcal{I}}) \leq (1 + 2\delta) \cdot \mathsf{cost}(\hat{\mathcal{S}}, \hat{\mathcal{I}})$.

Combining the three bounds proves Lemma D.1. □

Compared to Algorithm 5.2, to bound $\mathbb{E}[\mathsf{cost}(\mathcal{S}_{\mathrm{on}}, \mathcal{I})]$ with $\mathbb{E}[\mathsf{cost}(\mathcal{S}_{\mathrm{off}}, \hat{\mathcal{I}})]$, the only additional term we need to bound is $\mathsf{short}(\mathcal{S}_{\mathrm{on}}, \mathcal{I})$. The rest of the cost is identical. By analysis of Algorithm 5.2, for $\delta = c^{\frac{1}{5}}$, $(\mathbb{E}[\mathsf{cost}(\mathcal{S}_{\mathrm{on}}, \mathcal{I}) - \mathsf{short}(\mathcal{S}_{\mathrm{on}}, \mathcal{I})] - b)/\mathbb{E}[\mathsf{cost}(\mathcal{S}_{\mathrm{off}}, \hat{\mathcal{I}}) - \mathsf{short}(\mathcal{S}_{\mathrm{off}}, \hat{\mathcal{I}})] \leq (1 + \phi + c^{\frac{1}{5}})$. It then suffices to bound $\mathsf{short}[(\mathcal{S}_{\mathrm{on}}, \mathcal{I})]$, which we simply refer to as the short cost in the following analysis.

**Bad Collections.** To bound the short cost, we adopt a concept similar to that of a bad span.

**Definition D.2.** Fix some constant $\delta > 0$. A collection of spans $\mathcal{P} = \{P_1, \ldots, P_m\}$, where $m \leq (1/\delta) - 2$, is of type 1, 2, or 3 if there exists trips $\{(c_1, t_1), \ldots, (c_{m+1}, t_{m+1})\}$ such that $t_i + 1 < t_{i+1}$, and

$$\mathcal{P} = \{\mathcal{I}((t_i + 1, t_{i+1})) : 1 \leq i \leq m-1\} \cup \{\mathcal{I}((t_m + 1, t_{m+1} - 1))\},$$

and it respectively satisfies

1. There exists some $P_j \in \mathcal{P}$ such that $c(P_j) \geq (1+\delta)(1-\beta)^{-1}$

2. For all $P_j \in \mathcal{P}$, $c(P_j) < (1+\delta)(1-\beta)^{-1}$ and $c(\mathcal{P}) \geq \delta$

3. $c(\mathcal{P}) < \delta$

In other words, there exists a set of left-aligned cards whose gaps exactly correspond to spans in $\mathcal{P}$. Let the first trip in $P_1$ be $(c, t)$ and the last trip in $P_m$ be $(c', t')$. Fixing a sample $\hat{\mathcal{I}}$, we call a collection of spans $\mathcal{P}$ bad if $\hat{c}((t - \delta, t)) < 4(1 - \beta)^{-1}$ and $\hat{c}((t', t' + \delta)) < 4(1 - \beta)^{-1}$, and it satisfies one of the following:

1. $\mathcal{P}$ is of type 1, and for all $P_j \in \mathcal{P}$ such that $c(P_j) \geq (1 + \delta)(1 - \beta)^{-1}$, $\hat{c}(P_j) \leq (1 - \beta)^{-1}$;

2. $\mathcal{P}$ is of type 2, and $\hat{c}(\mathcal{P}) \leq c(\mathcal{P}) \cdot (1 + \delta)^{-1}$;

3. $\mathcal{P}$ is of type 3.

We say these bad collections are of type 1, 2, and 3, respectively. We call an interval between two long gaps of $\mathcal{S}_{\text{off}}$ bad if the spans of the short gaps in the interval are bad. We denote the number of bad collections of the three types in $\mathcal{S}_{\text{off}}$ by $\#\mathsf{badcol}_1$, $\#\mathsf{badcol}_2$, and $\#\mathsf{badcol}_3$, respectively.

We now compare $\mathsf{short}(\mathcal{S}_{\text{on}}, \mathcal{I})$ and $\mathsf{short}(\mathcal{S}_{\text{off}}, \hat{\mathcal{I}})$. Consider the collection of gaps $\mathcal{G}$ in a skip interval.

If $\mathcal{G}$ is not bad, it must be of type 2 by optimality of $\hat{\mathcal{S}}$. We therefore have $c(\mathcal{G}) \leq (1 + \delta) \cdot \hat{c}(\mathcal{G})$.

If $\mathcal{G}$ is bad of type 3, $c(\mathcal{G}) < \delta$. We may upper bound $\#\mathsf{badcol}_3$ by $\mathsf{buy}(\mathcal{S}_{\text{on}})$ since the number of skip intervals is upper bounded by the number of cards.

If $\mathcal{G}$ is bad of type 1 or 2, by the online adjustment of $\mathcal{S}_{\text{on}}$, every bad span incurs a cost of at most $(1 + 1/\delta)(1 + \delta)$. Therefore,

$$\mathsf{short}(\mathcal{S}_{\text{on}}, \mathcal{I}) \leq (1 + \delta) \cdot \mathsf{short}(\mathcal{S}_{\text{off}}, \hat{\mathcal{I}}) + \delta \cdot \#\mathsf{badcol}_3 + (1 + 1/\delta)(1 + \delta) \cdot (\#\mathsf{badcol}_1 + \#\mathsf{badcol}_2)$$
$$= (1 + \delta) \cdot \mathsf{short}(\mathcal{S}_{\text{off}}, \hat{\mathcal{I}}) + \delta \cdot \mathsf{buy}(\mathcal{S}_{\text{off}}) + (1 + 1/\delta)(1 + \delta) \cdot (\#\mathsf{badcol}_1 + \#\mathsf{badcol}_2)$$

Taking expectation with respect to the sampling, we now have

$$\mathbb{E}[\mathsf{short}(\mathcal{S}_{\text{on}}, \mathcal{I})] \leq (1 + \delta) \cdot \mathbb{E}[\mathsf{short}(\mathcal{S}_{\text{on}}, \hat{\mathcal{I}})] + \delta \cdot \mathsf{buy}(\mathcal{S}_{\text{on}}) + (1 + 1/\delta)(1 + \delta) \cdot (\mathbb{E}[\#\mathsf{badcol}_1] + \mathbb{E}[\#\mathsf{badcol}_2]). \quad (16)$$

It remains to bound $\mathbb{E}[\#\mathsf{badcol}_1]$ and $\mathbb{E}[\#\mathsf{badcol}_2]$. We again bound them by looking at looser quantities that are independent of the algorithm.

**Lemma D.3.** *If every trip $(c_i, t_i)$ satisfies $\alpha c \leq c_i \leq c$ for some constant $\alpha > 0$,*

$$\mathbb{E}[\#\mathsf{badcol}_2] \leq 2 \exp\left\{-\frac{2\delta^3 \varepsilon^2}{c(1 + \delta)^2}\right\} \cdot \frac{T}{\delta^3}$$
$$\cdot \left(\frac{4(1 + \delta)}{\alpha c(1 - \beta)} + \frac{1}{\alpha} + \left(\frac{8(1 + \delta)}{\alpha c(1 - \beta)} + \frac{2}{\alpha}\right) \cdot \left(1 - \exp\left\{-\frac{4(1 + \delta)\varepsilon^2}{c}\right\}\right)^{-1}\right)^2 \cdot \left(\frac{1 + \delta}{\alpha c(1 - \beta)}\right)^{\frac{1}{\delta}}.$$

*Proof of Lemma D.3.* Define $\#\mathsf{badcol}_2'$ to be the number of bad collections of spans in $\mathcal{I}$. Since every bad interval gives a bad collection of spans, we have $\#\mathsf{badcol}_2 \leq \#\mathsf{badcol}_2'$. It then suffices to bound $\mathbb{E}[\#\mathsf{badcol}_2']$. By definition,

$$\mathbb{E}[\#\mathsf{badcol}_2'] = \sum_{\text{type 2 collection } \mathcal{P}} \mathbb{P}(\mathcal{P} \text{ is bad}).$$

Let the collection $\mathcal{P}$ start at $(c, t)$ and end at $(c', t')$. Recall that for $\mathcal{P}$ to be bad of type 2, the following three should hold: $\hat{c}(\mathcal{P}) \leq c(\mathcal{P})/(1 + \delta)$, $\hat{c}((t - \delta, t)) < 4$, and $\hat{c}((t', t' + \delta)) < 4$. Since the sampling is independent on disjoint sets of trips, we bound these probabilities separately. First, we bound the probability that $\hat{c}(\mathcal{P}) \leq c(\mathcal{P})/(1 + \delta)$. Since $c(\mathcal{P}) \geq \delta$, the

probability can be bounded by Hoeffding's inequality (Theorem A.1):

$$
\begin{aligned}
\mathbb{P}\left[\hat{c}(\mathcal{P}) \le \frac{c(\mathcal{P})}{1+\delta}\right] &\le \mathbb{P}\left[|\hat{c}(\mathcal{P}) - c(\mathcal{P})| \ge c(\mathcal{P}) \cdot \left(\frac{\delta}{1+\delta}\right)\right] \\
&\le 2\exp\left\{-\frac{2\left(\frac{\delta}{1+\delta}\right)^2 \cdot c(\mathcal{P})^2}{\sum_{(c_i,t_i)\in\mathcal{P}} \frac{c_i^2}{\varepsilon^2}}\right\} \\
&\le 2\exp\left\{-\frac{2\left(\frac{\delta}{1+\delta}\right)^2 \cdot c(\mathcal{P})^2 \cdot \varepsilon^2}{c \cdot c(\mathcal{P})}\right\} \\
&\le 2\exp\left\{-\frac{2\delta^3\varepsilon^2}{c(1+\delta)^2}\right\}
\end{aligned}
$$

So, we have

$$
\mathbb{E}[\#\mathsf{badcol}_2'] \le 2\exp\left\{-\frac{2\delta^3\varepsilon^2}{c(1+\delta)^2}\right\} \cdot \sum_{\text{type 2 collection } \mathcal{P}} \mathbb{P}\left[\hat{c}((t-\delta,t)) < \frac{4}{1-\beta}\right] \cdot \mathbb{P}\left[\hat{c}((t',t'+\delta)) < \frac{4}{1-\beta}\right].
$$

To bound the sum of the probabilities above, count the number of type 2 collections $\mathcal{P}$ by counting the number of starting and ending trips, and then counting the number of collections of spans of type 2 in the time interval fixed by the two endpoints. We therefore have

$$
\begin{aligned}
&\sum_{\text{type 2 collection } \mathcal{P}} \mathbb{P}\left[\hat{c}((t-\delta,t)) < \frac{4}{1-\beta}\right] \cdot \mathbb{P}\left[\hat{c}((t',t'+\delta)) < \frac{4}{1-\beta}\right] \\
&= \sum_{t\in[0,T]} \mathbb{P}\left[\hat{c}((t-\delta,t)) < \frac{4}{1-\beta}\right] \sum_{t'\in[t+1,t+\frac{1}{\delta})} \mathbb{P}\left[\hat{c}((t',t'+\delta)) < \frac{4}{1-\beta}\right]
\end{aligned}
$$

$$
[\text{\# type 2 collections that start at } t \text{ and end at } t'].
$$

We first bound

$$
\sum_{t\in[0,T]} \mathbb{P}\left[\hat{c}((t-\delta,t)) < \frac{4}{1-\beta}\right].
$$

Evenly partition the original instance $I$ into time intervals of length $\delta$ to get $J_1,\ldots,J_{T/\delta}$. Separately consider the collections $\mathcal{P}$ that start in one such interval $J_i$. Then,

$$
\sum_{t\in[0,T]} \mathbb{P}\left[\hat{c}((t-\delta,t)) < \frac{4}{1-\beta}\right] = \sum_{i=1}^{T/\delta}\sum_{t\in J_i} \mathbb{P}\left[\hat{c}((t-\delta,t)) < \frac{4}{1-\beta}\right].
$$

We now separately bound each

$$
\sum_{t\in J_i} \mathbb{P}\left[\hat{c}((t-\delta,t)) < \frac{4}{1-\beta}\right].
$$

Greedily partition $J_i$ into spans $J_{i,1},\ldots,J_{i,n}$, such that $4(1+\delta)/(1-\beta) \le c(J_{i,k}) \le 4(1+\delta)/(1-\beta)+c$ for $1\le k\le n-1$ and $c(J_{i,n}) \le 4(1+\delta)/(1-\beta)+c$. If $c(J_i) \le 4(1+\delta)/(1-\beta)$, just take $J_{i,1}=J_i$. For $t\in J_{i,k}$ where $k>1$, since

$$
c(J_{i,1}\cup\cdots\cup J_{i,k-1}) \ge \frac{4(k-1)(1+\delta)}{1-\beta},
$$

by Hoeffding's inequality, we have

$$\mathbb{P}\left[\hat{c}((t-\delta,t)) < \frac{4}{1-\beta}\right] \le \mathbb{P}\left[\hat{c}((t-\delta,t)) < \frac{c((t-\delta,t))}{(k-1)(1+\delta)}\right]$$

$$\le \mathbb{P}\left[|\hat{c}((t-\delta,t)) - c((t-\delta,t))| > \frac{k-2}{k-1} \cdot c((t-\delta,t))\right]$$

$$\le 2\exp\left\{-\frac{2 \cdot \left(\frac{k-2}{k-1}\right)^2 \cdot c((t-\delta,t))^2}{\sum_{t_i \in (t-\delta,t)} \frac{c_i^2}{\varepsilon^2}}\right\}$$

$$\le 2\exp\left\{-\frac{2 \cdot \left(\frac{k-2}{k-1}\right)^2 \cdot c((t-\delta,t))^2 \cdot \varepsilon^2}{c \cdot c((t-\delta,t))}\right\}$$

$$\le 2\exp\left\{-\frac{4(k-2)(1+\delta)\varepsilon^2}{c(1-\beta)}\right\}$$

Therefore,

$$\sum_{t \in J_i} \mathbb{P}\left[\hat{c}((t-\delta,t)) < \frac{4}{1-\beta}\right] = \sum_{t \in J_{i,1}} \mathbb{P}\left[\hat{c}((t-\delta,t)) < \frac{4}{1-\beta}\right] + \sum_{k=2}^{n}\sum_{t \in J_{i,k}} \mathbb{P}\left[\hat{c}((t-\delta,t)) < \frac{4}{1-\beta}\right]$$

$$\le \left(\frac{4(1+\delta)}{\alpha c(1-\beta)} + \frac{1}{\alpha}\right) + \left(\frac{4(1+\delta)}{\alpha c(1-\beta)} + \frac{1}{\alpha}\right) \cdot \sum_{k=2}^{n} 2\exp\left\{-\frac{4(k-2)(1+\delta)\varepsilon^2}{c}\right\}$$

$$\le \left(\frac{4(1+\delta)}{\alpha c(1-\beta)} + \frac{1}{\alpha}\right) + \left(\frac{8(1+\delta)}{\alpha c(1-\beta)} + \frac{2}{\alpha}\right) \cdot \left(1 - \exp\left\{-\frac{4(1+\delta)\varepsilon^2}{c}\right\}\right)^{-1}$$

This gives the bound

$$\sum_{t \in [0,T]} \mathbb{P}\left[\hat{c}((t-\delta,t)) < \frac{4}{1-\beta}\right] = \sum_{i=1}^{T/\delta}\sum_{t \in J_i} \mathbb{P}\left[\hat{c}((t-\delta,t)) < \frac{4}{1-\beta}\right]$$

$$\le \frac{T}{\delta} \cdot \left(\frac{4(1+\delta)}{\alpha c(1-\beta)} + \frac{1}{\alpha} + \left(\frac{8(1+\delta)}{\alpha c(1-\beta)} + \frac{2}{\alpha}\right) \cdot \left(1 - \exp\left\{-\frac{4(1+\delta)\varepsilon^2}{c}\right\}\right)^{-1}\right).$$

We now bound

$$\sum_{t' \in [t+1,t+\frac{1}{\delta})} \mathbb{P}\left[\hat{c}((t',t'+\delta)) < \frac{4}{1-\beta}\right].$$

Note that this is entirely the same as

$$\sum_{t \in [0,T]} \mathbb{P}\left[\hat{c}((t-\delta,t)) < \frac{4}{1-\beta}\right],$$

but with $T = 1/\delta$. We therefore have the bound

$$\sum_{j=1}^{1/\delta}\sum_{t' \in [t+1,t+\frac{1}{\delta})} \mathbb{P}\left[\hat{c}((t',t'+\delta)) < \frac{4}{1-\beta}\right] \le \frac{1}{\delta^2} \cdot \left(\frac{4(1+\delta)}{\alpha c(1-\beta) + \frac{1}{\alpha}} + \left(\frac{8(1+\delta)}{\alpha c(1-\beta)} + \frac{2}{\alpha}\right) \cdot \left(1 - \exp\left\{-\frac{4(1+\delta)\varepsilon^2}{c}\right\}\right)^{-1}\right).$$

Now, we bound [# type 2 collections that start at $t$ and end at $t'$] for fixed $t$ and $t'$. Since each such collection correspond to the gaps of a certain set of cards, we just count the number of configurations of cards. Since at most $1/\delta$ cards are bought, the quantity can be bounded as follows:

[# type 2 collections that start at $t$ and end at $t'$]

$$\le \prod_{i=1}^{1/\delta} [\text{number of positions for card } i \text{ when card } 1,\dots,i-1 \text{ are fixed}].$$

In a type 2 collection, every gap has cost $\leq (1 + \delta)/(1 - \beta)$, so

$$[\text{number of positions for card } i \text{ when card } 1, \ldots, i-1 \text{ are fixed}] \leq \frac{1 + \delta}{\alpha c(1 - \beta)}.$$

Therefore, we have that

$$[\text{\# type 2 collections that start at } t \text{ and end at } t'] \leq \left( \frac{1 + \delta}{\alpha c(1 - \beta)} \right)^{\frac{1}{\delta}}.$$

Combining these bounds above gives the following bound for $\mathbb{E}[\#\mathsf{badcol}'_2]$:

$$\mathbb{E}[\#\mathsf{badcol}'_2] \leq 2 \exp \left\{ -\frac{2\delta^3 \varepsilon^2}{c(1 + \delta)^2} \right\} \cdot \frac{T}{\delta^3}$$

$$\cdot \left( \frac{4(1 + \delta)}{\alpha c(1 - \beta)} + \frac{1}{\alpha} + \left( \frac{8(1 + \delta)}{\alpha c(1 - \beta)} + \frac{2}{\alpha} \right) \cdot \left( 1 - \exp \left\{ -\frac{4(1 + \delta)\varepsilon^2}{c} \right\} \right)^{-1} \right)^2 \cdot \left( \frac{1 + \delta}{\alpha c(1 - \beta)} \right)^{\frac{1}{\delta}} \quad (17)$$

Lemma D.3 then follows from $\#\mathsf{badcol}_2 \leq \#\mathsf{badcol}'_2$. $\qquad\qquad\square$

**Lemma D.4.** *If every trip $(c_i, t_i)$ satisfies $\alpha c \leq c_i \leq c$ for some constant $\alpha > 0$,*

$$\mathbb{E}[\#\mathsf{badcol}_1] \leq \frac{T}{\delta^3} \cdot \left( \frac{4(1 + \delta)}{\alpha c(1 - \beta)} + \frac{1}{\alpha} + \left( \frac{8(1 + \delta)}{\alpha c(1 - \beta)} + \frac{2}{\alpha} \right) \cdot \left( 1 - \exp \left\{ -\frac{4(1 + \delta)\varepsilon^2}{c} \right\} \right)^{-1} \right)^2$$

$$\cdot \left( \frac{1 + \delta}{\alpha c(1 - \beta)} \right)^{\frac{1}{\delta}} \cdot 2 \exp \left\{ -\frac{2\delta^2 \varepsilon^2}{c(1 + \delta)} \right\}.$$

*Proof.* Define $\#\mathsf{badcol}'_1$ to be the size of the largest family of disjoint bad collections of type 1 in $\mathcal{I}$. Two bad collections of spans $\mathcal{P}_1$ and $\mathcal{P}_2$ are disjoint if all trips in $\mathcal{P}_1$ are earlier than all trips in $\mathcal{P}_2$ (or the other way around). Since bad collections of spans that arise from short gaps of $\mathcal{S}$ are disjoint, we have $\#\mathsf{badcol}_1 \leq \#\mathsf{badcol}'_1$. It then suffices to bound $\mathbb{E}[\#\mathsf{badcol}'_1]$. Note that for a fixed choice of starting trip and ending trip, the largest family of disjoint bad collections contains at most one collection with the given start and end trips. Therefore, using the same counting from Lemma D.3, we have

$$\mathbb{E}[\#\mathsf{badcol}'_1] \leq \sum_{t \in [0,T]} \mathbb{P}\left[ \hat{c}((t - \delta, t)) < \frac{4}{1 - \beta} \right] \sum_{t' \in [t+1, t+\frac{1}{\delta})} \mathbb{P}\left[ \hat{c}((t', t' + \delta)) < \frac{4}{1 - \beta} \right]$$

$$\mathbb{P}\left[\text{there exists bad } \mathcal{P} \text{ that starts at } t \text{ and ends at } t'\right]$$

The first two terms are bounded as in Lemma D.3.

$$\sum_{t \in [0,T]} \mathbb{P}\left[ \hat{c}((t - \delta, t)) < \frac{4}{1 - \beta} \right] \sum_{t' \in [t+1, t+\frac{1}{\delta})} \mathbb{P}\left[ \hat{c}((t', t' + \delta)) < \frac{4}{1 - \beta} \right]$$

$$\leq \frac{T}{\delta^3} \cdot \left( \frac{4(1 + \delta)}{\alpha c(1 - \beta) + \frac{1}{\alpha}} + \left( \frac{8(1 + \delta)}{\alpha c(1 - \beta)} + \frac{2}{\alpha} \right) \cdot \left( 1 - \exp \left\{ -\frac{4(1 + \delta)\varepsilon^2}{c} \right\} \right)^{-1} \right)^2$$

It suffices to bound the last term.

Categorize bad collections $\mathcal{P} = \{P_1, \ldots, P_m\}$ by the first $P_k$ such that $c(P_i) \geq 1 + \delta$, and we say $\mathcal{P}$ is of category $k$ in this case. By definition, for $\mathcal{P}$ of category $k$, $c(P_i) < 1 + \delta$ for all $i \leq k - 1$, and $c(P_k) \geq 1 + \delta$. Therefore, to bound the probability that there exists bad $\mathcal{P}$ of category $k$, note that

$$\mathbb{P}\left[\text{there exists bad } \mathcal{P} \text{ of category } k \text{ which starts at } t \text{ and ends at } t'\right]$$

$$\leq [\text{\#card positions of the first } k - 1 \text{ cards such that } c(P_i) < 1 + \delta \text{ for } i \leq k - 1] \cdot \mathbb{P}\left[ \hat{c}(P_k) \leq \frac{1}{1 - \beta} \right]$$

The first term can be bounded the same way as in Lemma D.3:

$$[\text{\#card positions of the first } k-1 \text{ cards such that } c(P_i) < 1 + \delta \text{ for } i \leq k-1] \leq \left(\frac{1+\delta}{\alpha c \cdot (1-\beta)}\right)^{\frac{1}{\delta}}.$$

To bound the probability that $\hat{c}(P_k) \leq 1/(1-\beta)$ for every $P_k$, use Hoeffding's inequality:

$$\mathbb{P}\left[\hat{c}(P_k) \leq \frac{c(P_k)}{1+\delta}\right] \leq \mathbb{P}\left[|\hat{c}(P_k) - c(P_k)| \geq c(P_k) \cdot \left(\frac{\delta}{1+\delta}\right)\right]$$

$$\leq 2\exp\left\{-\frac{2\left(\frac{\delta}{1+\delta}\right)^2 \cdot c(P_k)^2}{\sum_{(c_i,t_i)\in P_k}\frac{c_i^2}{\varepsilon^2}}\right\}$$

$$\leq 2\exp\left\{-\frac{2\left(\frac{\delta}{1+\delta}\right)^2 \cdot c(P_k)^2 \cdot \varepsilon^2}{c \cdot c(P_k)}\right\}$$

$$\leq 2\exp\left\{-\frac{2\delta^2\varepsilon^2}{c(1+\delta)}\right\}$$

Combining all the bounds, we have

$$\mathbb{E}[\text{\#badcol}'_1] \leq \frac{T}{\delta^3} \cdot \left(\frac{4(1+\delta)}{\alpha c(1-\beta)} + \frac{1}{\alpha} + \left(\frac{8(1+\delta)}{\alpha c(1-\beta)} + \frac{2}{\alpha}\right) \cdot \left(1 - \exp\left\{-\frac{4(1+\delta)\varepsilon^2}{c}\right\}\right)^{-1}\right)^2$$

$$\cdot \left(\frac{1+\delta}{\alpha c(1-\beta)}\right)^{\frac{1}{\delta}} \cdot 2\exp\left\{-\frac{2\delta^2\varepsilon^2}{c(1+\delta)}\right\} \quad (18)$$

Lemma D.4 then follows from $\text{\#badcol}_1 \leq \text{\#badcol}'_1$. $\qquad\square$

Note that on the original instance, we may assume that there is at least one trip in every unit time interval, as otherwise the instance can be partitioned into two separate instances. Therefore, we may assume $\mathbb{E}[\text{cost}(\mathcal{S},\mathcal{I})] \geq \alpha \cdot c \cdot T$.

We now choose proper $\delta$ for the algorithm.

**Lemma D.5.** *For $\delta = c^{\frac{1}{5}}$,*

$$\frac{\mathbb{E}[\text{\#badcol}_2]}{\mathbb{E}[\text{cost}(\mathcal{S},\mathcal{I})]} \leq 2\exp\left\{-\frac{2\varepsilon^2}{c^{\frac{2}{5}}(1+\delta)^2}\right\} \cdot \frac{1}{\alpha c^{\frac{8}{5}}}$$

$$\cdot \left(\frac{4(1+c^{\frac{1}{5}})}{\alpha c(1-\beta)} + \frac{1}{\alpha} + \left(\frac{8(1+c^{\frac{1}{5}})}{\alpha c(1-\beta)} + \frac{2}{\alpha}\right) \cdot \left(1 - \exp\left\{-\frac{4(1+c^{\frac{1}{5}})\varepsilon^2}{c}\right\}\right)^{-1}\right)^2 \cdot \left(\frac{1+c^{\frac{1}{5}}}{\alpha c(1-\beta)}\right)^{c^{-\frac{1}{5}}} =: \phi'_2$$

*and*

$$\frac{\mathbb{E}[\text{\#badcol}_1]}{\mathbb{E}[\text{cost}(\mathcal{S},\mathcal{I})]} \leq \frac{1}{\alpha c^{\frac{8}{5}}} \cdot \left(\frac{4(1+c^{\frac{1}{5}})}{\alpha c(1-\beta)} + \frac{1}{\alpha} + \left(\frac{8(1+c^{\frac{1}{5}})}{\alpha c(1-\beta)} + \frac{2}{\alpha}\right) \cdot \left(1 - \exp\left\{-\frac{4(1+c^{\frac{1}{5}})\varepsilon^2}{c}\right\}\right)^{-1}\right)^2$$

$$\cdot \left(\frac{1+c^{\frac{1}{5}}}{\alpha c(1-\beta)}\right)^{c^{-\frac{1}{5}}} \cdot 2\exp\left\{-\frac{2\varepsilon^2}{c^{\frac{3}{5}}(1+c^{\frac{1}{5}})}\right\} =: \phi'_1.$$

*In particular, $\phi'_1 \to 0$ and $\phi'_2 \to 0$ as $c \to 0$.*

*Proof.* Immediately follows from Lemma D.3, Lemma D.4, and the assumption that $\mathbb{E}[\text{cost}(\mathcal{S},\mathcal{I})] \geq \alpha cT$. $\qquad\square$

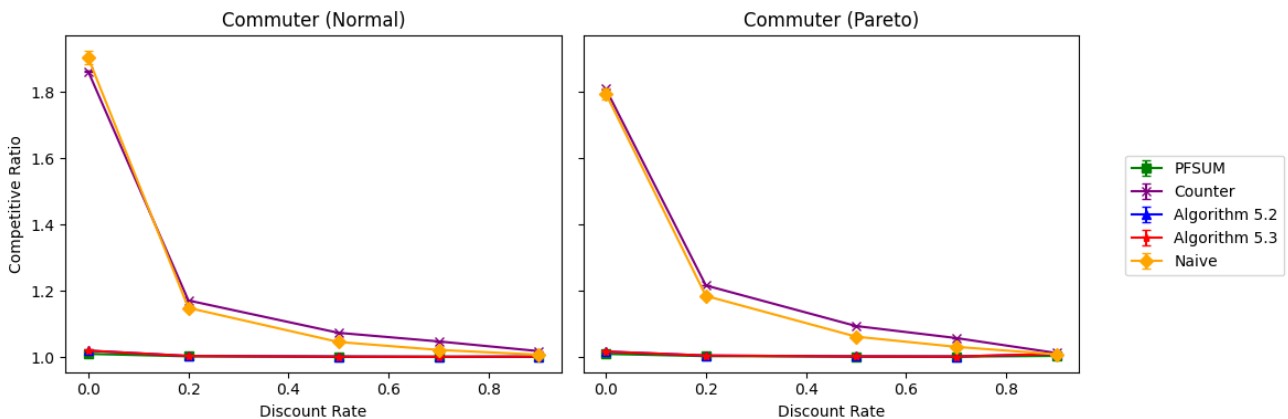

*Figure 3.* The cost ratios of the algorithms on commuter instances (sampling rate $\varepsilon = 0.1$).

**Lemma D.6** (Theorem 5.4 restated). *Let $\phi' = \phi'_1 + \phi'_2$. By setting $\delta = c^{\frac{1}{5}}$, the competitive ratio of Algorithm 5.3 is bounded by $\max\{1 + \phi + c^{\frac{1}{5}}, 1 + 2c^{\frac{1}{5}} + (1 + (1/c^{\frac{1}{5}}))(1 + c^{\frac{1}{5}})\phi'\}$. In particular, the competitive ratio of Algorithm 5.3 approaches $1$ as $c \to 0$.*

*Proof.* Follows from Equation 16, Lemma D.5, and Theorem 5.1. $\square$

## E. Experimental results

Figures 3, 4, 5 and 6 show performance across different arrival patterns under a low sampling rate $\varepsilon = 0.1$, while 7, 8, 9 and 10 present results under the same setups with noisy samples.

In the low-sampling regime, Algorithms 4.2 and 4.3 already achieve strong performance on several instance classes. For commuter arrivals and occasional arrivals with mean inter-arrival time 2, both of our algorithms obtain near-optimal competitive ratios, both with and without noise, comparable to those of PFSUM.

On cluster instances, all algorithms have high cost ratio, especially with noise in the sample. This is a consequence of the inherent brittleness of the optimal solution on cluster instances: even if a small number of trips are introduced to a gap between two clusters, or missing trips near the boundary of a cluster in the sample, can substantially change the optimal solution. In the noiseless setting, our algorithms outperform the PFSUM, but in the absence of theoretical robustness guarantees, their performance degrades under noisy sampling.

For occasional arrivals with mean inter-arrival time 40, our algorithms underperform PFSUM. In this sparse regime, there are on average only five trips within the duration of a card, and combined with a low sampling rate, the sample provides very limited information. This behavior is therefore consistent with expectations and highlights a regime where PFSUM is better suited.

Figures 11, 12, 13 and 14 present results in high-sampling regime with noisy samples.

Under noisy sampling, our algorithms maintain similar performance on commuter instances and occasional instances with mean inter-arrival time 2. On occasional instances with mean inter-arrival time 40 and on clustered instances, although we observe modest improvement compared to the low-sampling regime, our algorithms do not yet achieve the level of consistency exhibited by PFSUM.

We finally note that Algorithms 5.2 and 5.3 demonstrate identical performance across all tested inputs, which is not a coincidence. Based on the results of Section 5, we expect Algorithm 5.3 to behave more similarly to the naïve sampling algorithm in regimes where the latter outperforms other algorithms. However, the theoretical guarantees in Section 5 hold only in the limit. To observe the predicted separation empirically, the ratio between card cost and individual trip costs would need to be substantially larger than in our current experimental setting. Due to runtime constraints, we do not explore this extreme regime, and therefore do not observe a measurable difference between Algorithms 4.2 and 4.3 in our experiments.

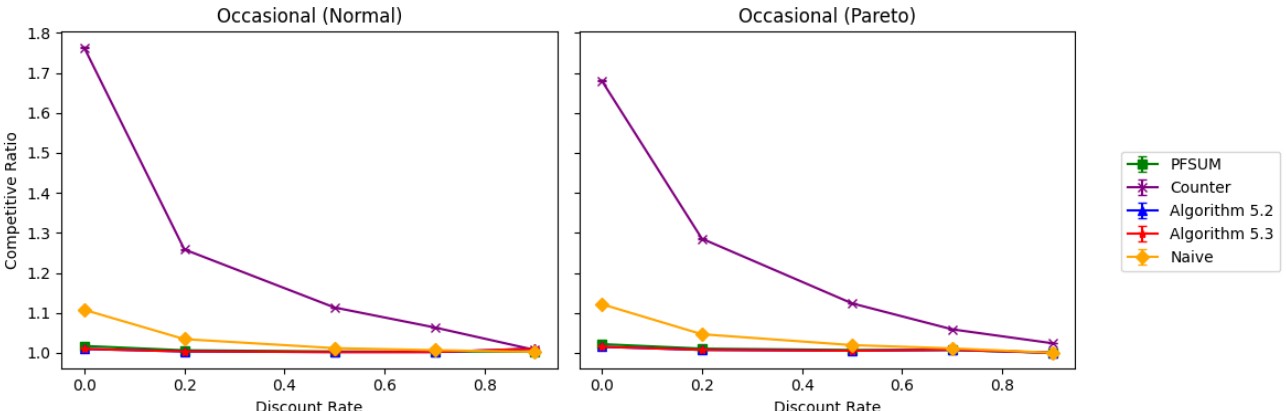

*Figure 4.* The cost ratios of the algorithms on occasional instances (mean interval length 2, sampling rate $\varepsilon = 0.1$).

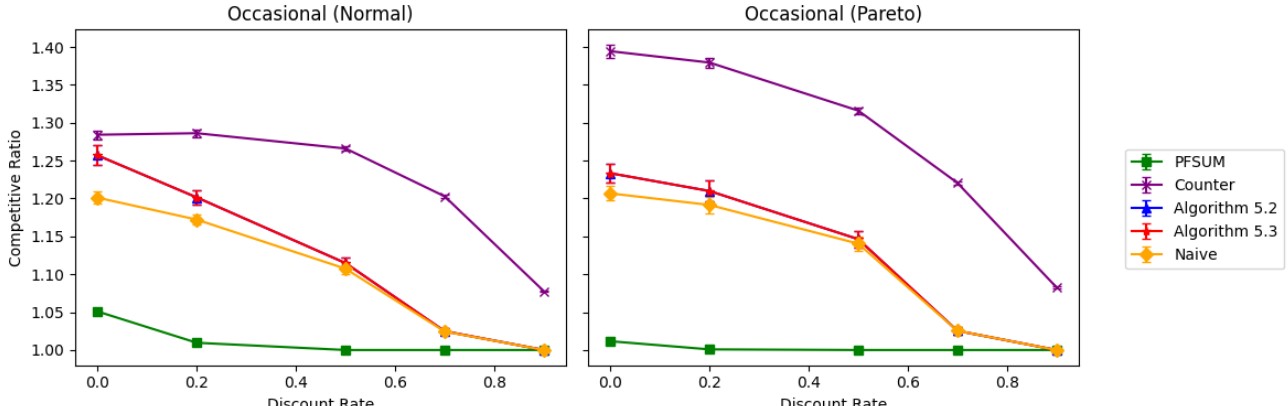

*Figure 5.* The cost ratios of the algorithms on occasional instances (mean interval length 40, sampling rate $\varepsilon = 0.1$).

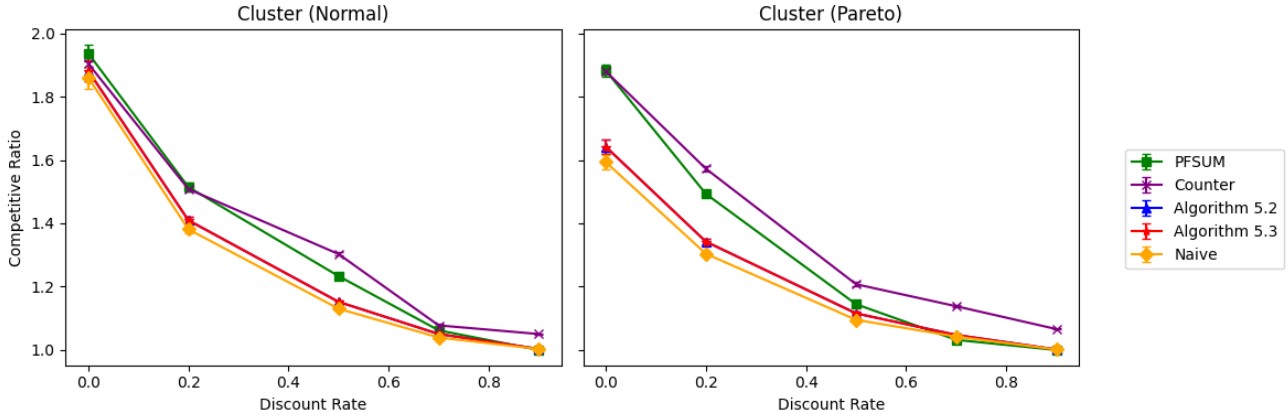

*Figure 6.* The cost ratios of the algorithms on cluster instances (sampling rate $\varepsilon = 0.1$).

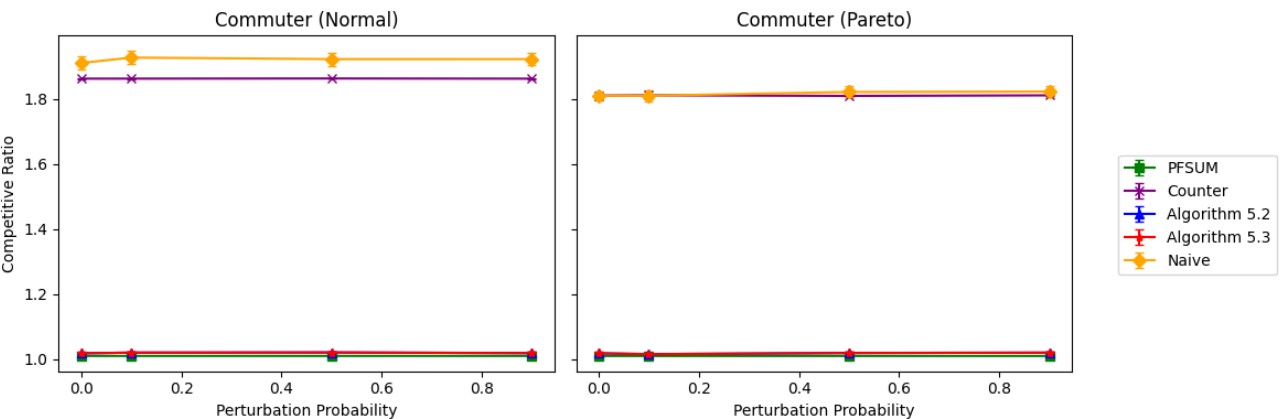

*Figure 7.* The cost ratios of the algorithms on commuter instances with noisy samples (sampling rate $\varepsilon = 0.1$, discount rate $\beta = 0$).

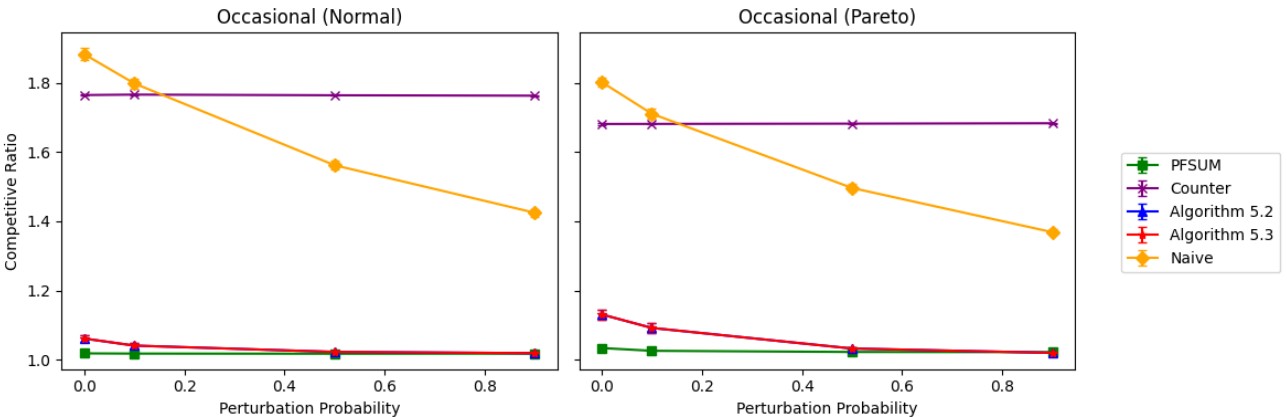

*Figure 8.* The cost ratios of the algorithms on occasional instances with noisy samples (mean interval length 2, sampling rate $\varepsilon = 0.1$, discount rate $\beta = 0$).

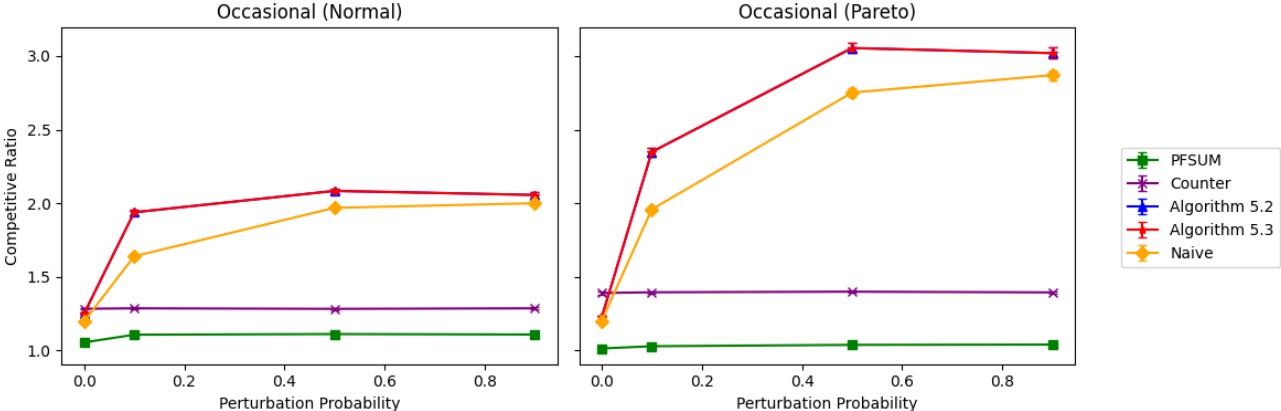

*Figure 9.* The cost ratios of the algorithms on occasional instances with noisy samples (mean interval length 40, sampling rate $\varepsilon = 0.1$, discount rate $\beta = 0$).

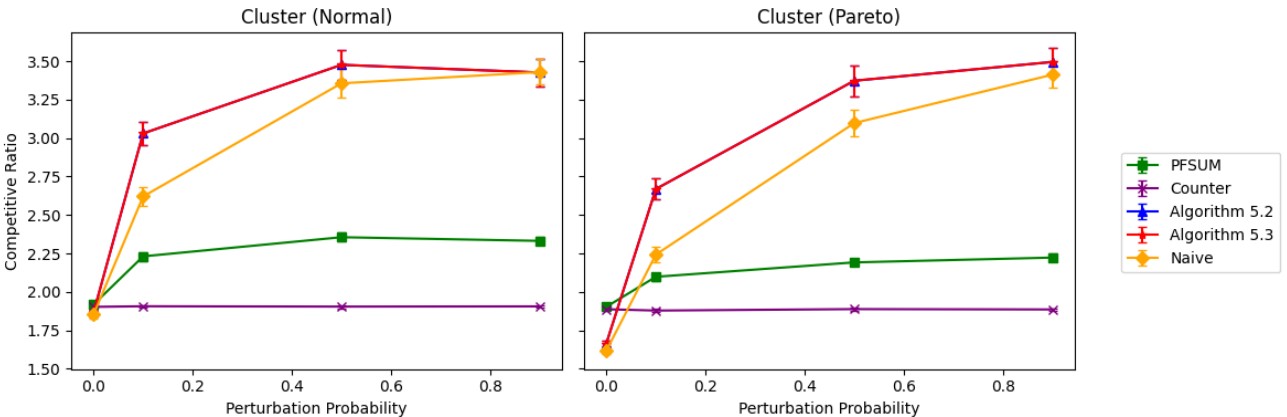

*Figure 10.* The cost ratios of the algorithms on cluster instances with noisy samples (sampling rate $\varepsilon = 0.1$, discount rate $\beta = 0$).

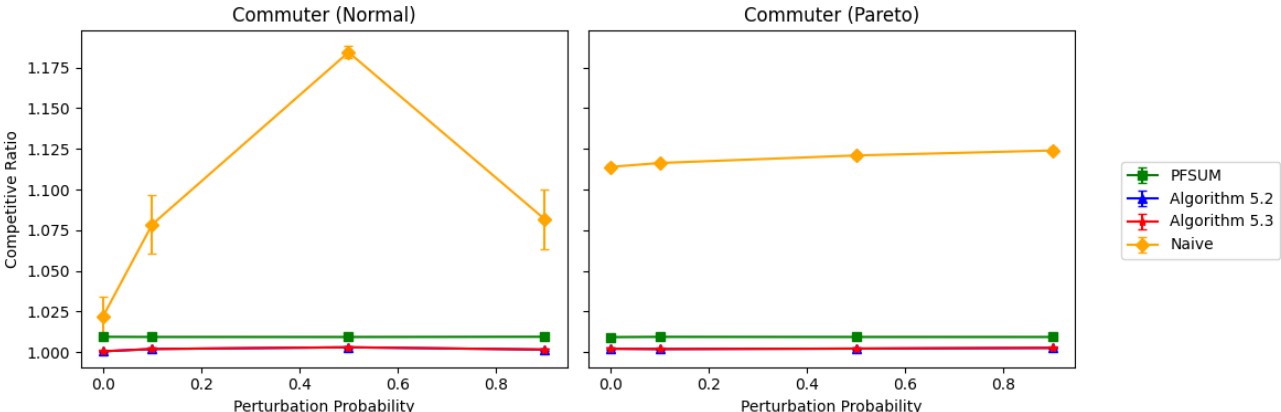

*Figure 11.* The cost ratios of the algorithms on commuter instances with noisy samples (sampling rate $\varepsilon = 0.5$, discount rate $\beta = 0$).

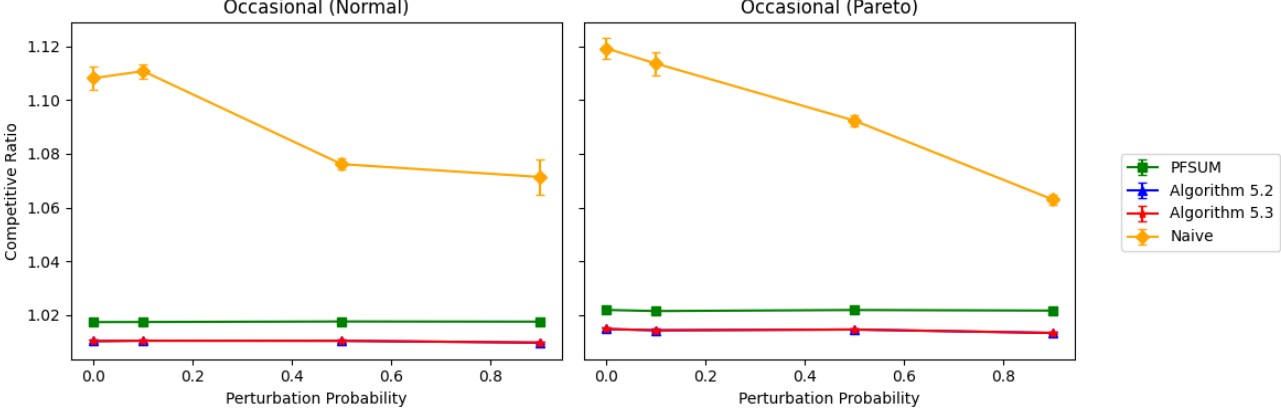

*Figure 12.* The cost ratios of the algorithms on occasional instances with noisy samples (mean interval length 2, sampling rate $\varepsilon = 0.5$, discount rate $\beta = 0$).

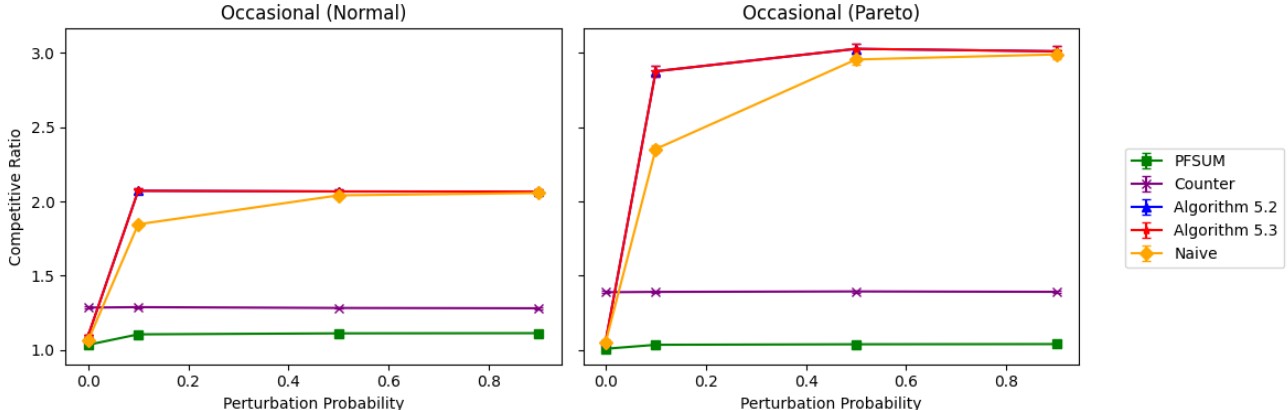

*Figure 13.* The cost ratios of the algorithms on occasional instances with noisy samples (mean interval length 40, sampling rate $\varepsilon = 0.5$, discount rate $\beta = 0$).

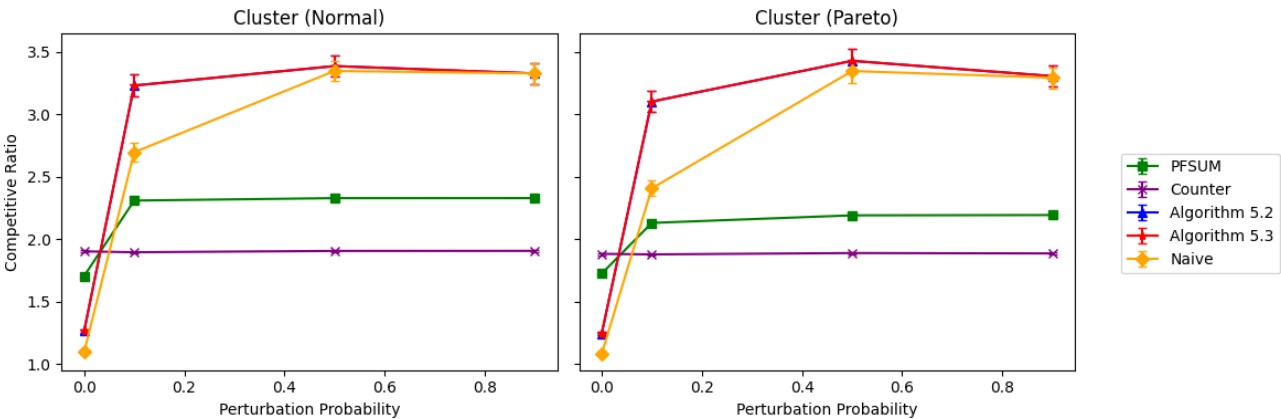

*Figure 14.* The cost ratios of the algorithms on cluster instances with noisy samples (sampling rate $\varepsilon = 0.5$, discount rate $\beta = 0$).

