# OpenReview forum: "Learning-augmented Rent-or-Buy with a Sample"
_ICML.cc/2026/Conference — ICML 2026 regular_

### Official Review · Reviewer_sDM6 · 2026-02-25

**Soundness:** 4
**Presentation:** 2
**Significance:** 3
**Originality:** 3
**Overall Recommendation:** 4
**Confidence:** 4

**Summary:**

The paper studied the classical online decision-making problem of rent-or-buy (the Bahncard problem). In this problem, we are given $T$ total days and $n$ possible trips, and each trip will incur a cost $c_i$. The decision maker has the option of paying the full cost of $c_i$ or ‘buying a card’ with cost $1$ to cover the costs (potentially) occurring in $[t, t+1]$. The goal is to come up with a strategy to minimize the cost, and in particular, to be competitive with the optimal strategy if we know when the trip will occur from the beginning.

The paper studied the problem in the setting where a *sample* of the trip is available, i.e., for each trip ${t_i, c_i}$ that encodes the time and cost, the information is revealed to the decision maker with probability $\varepsilon$. The main results include:
- When the ratio of trip cost and card cost goes to $0$, they designed an algorithm with a competitive ratio $3/2$, improving the previous best bound of $2$.
- Assuming all trips are in range $[\alpha c, c]$, the competitive ratio approaches $1$.

*Main techniques.** The paper focused on analyzing the additional information we can get from the sample of the trip. For the special case of ski rental (prices are uniform, and the card discount is 1), the paper designed a very simple algorithm by simply following the optimal solution on the sampled trip. For the more general Bahncard problem, the paper first identified a reason for the naive algorithm to fail: i.e., the sampled days contain gap between trips, and not buying cards during these days would lead to higher costs. Therefore, the paper designed an algorithm that buys cards to cover the trips between the ‘long gaps’, which yields a competitive ratio of 3/2.

**Compliance With Llm Reviewing Policy:**

Affirmed.

**Final Justification:**

I believe my concerns have all been resolved, especially for the possible scenario of false positive signals. Overall, I remain inclined to acceptance.

**Key Questions For Authors:**

See weakness comments. Some MISC comments:
- The notation of competitive ratio $\phi$ was defined in Theorem 4.1, but is overloaded as a parameter to denote something else in Theorem 5.1.
- In section 5, when you’re constructing the hard instance for the naive algorithm, consider not using parameter $n$ – this is overloaded for the total number of trips.

**Limitations:**

Yes (No foreseeable societal negative impact)

**Strengths And Weaknesses:**

**Strength**

I like the setting of the learning-augmented algorithm. Compared to related papers in the literature that use predictions for involved parameters (e.g., the dual parameter), the prediction for ‘a sample of the trips’ is much more natural and easy to obtain in practice. The paper gave enough intuitions for the ski rental algorithm, and the more involved algorithm, and the reason why the more involved algorithm is needed is clear. Finally, the paper also ran numerical simulations to verify the performance of their algorithm.

**Weakness**

Although the prediction is much more natural than, e.g., predicting some parameters of certain algorithms, the learning-augmented model does not account for the *false positive* of the trips. It appears to me that their algorithm will fall apart if there are ‘false positive’ days that are predicted for a trip to happen, but there is actually no trip.

Another criticism is the way the paper presented the theorems. I think the statements focus too much on the asymptotic behavior and completely ignore the dependency on the parameters, e.g., the dependency on $\varepsilon$. I briefly checked the details in the appendix, and I understand that maybe the reason is that the parameters are too involved. Nevertheless, I think the paper will benefit from giving more detailed bounds with the parameter dependencies.

---

> ### Author Rebuttal · Authors · 2026-03-30
>
> We thank the reviewer for their insightful comments and positive evaluation of our work. We address their concerns below.
>
> Weakness 1
>
> Our framework assumes accurate but sparse samples, emphasizing limited information rather than noisy predictions. We are aware of one paper (Zhao, Tang, Chen, and Deng, Neurips 24) that uses possibly noisy predictions for the Bahncard problem. However, this work does not achieve any improvement over the worst-case bounds if the discount factor $\beta = 0$. (Note that as mentioned in our paper, $\beta = 0$ is the most challenging setting, since the algorithm’s choices determine the largest share of the cost in this case.) In contrast, we show that a sparse sample of the input can improve over worst-case performance, even for the case of $\beta = 0$ which subsumes all other values of $\beta$.
>
> We also note that we empirically evaluate the performance of our algorithms in a noisy setting where the sample can have errors (see Section 6 and Appendix E). More precisely, we first add noise to the original instance, then sample from this noisy instance, and use these samples to guide the algorithm on the original instance. The noisy instance is created from the original instance by randomly adding false negatives and false positives: we randomly delete some existing trips and add some non-existent ones. This methodology follows prior work by Zhao et al. in Neurips 24. The experimental results (see Figures 7-14 in Appendix E) demonstrate that our algorithms remain robust to sampling errors across several natural families of instances.
>
> Weakness 2
>
> The performance guarantees in their exact form are stated in the appendix. Due to the length and complexity of the expressions, they are abbreviated to asymptotic forms in the main body of the paper for easier comprehension. We will add the precise forms of the expression to the main body of the paper in the final version, as suggested by the reviewer.
>
> Questions 1 and 2
>
> Thank you for pointing out the notational inconsistencies. We will fix these typos in the final version of our paper.
>
> We hope these clarifications address the reviewer’s concerns. We are happy to provide further details.

---

> > ### Author Rebuttal · Reviewer_sDM6 · 2026-04-04
> >
> > Thanks for getting back to me about my review comments. I believe my concerns have all been resolved, especially for the possible scenario of false positive signals. Overall, I remain inclined to acceptance.

---

> > > ### Author Response · Authors · 2026-04-04
> > >
> > > Thank you for your positive response to our rebuttal comments. We truly appreciate it.

---

### Official Review · Reviewer_kzr2 · 2026-03-05

**Soundness:** 4
**Presentation:** 4
**Significance:** 3
**Originality:** 3
**Overall Recommendation:** 5
**Confidence:** 4

**Summary:**

The paper studies the Bahncard problem in the framework of algorithms with predictions. In the classic version of the problem, a sequence of trips, each with an associated cost, arrives online. At any time, a discount card can be purchased for a fixed cost $C$, which reduces the cost of trips occurring in the following time window of fixed length $\Delta$ to a $\beta$ fraction of their original cost. This generalizes the ski rental problem, and the objective is to minimize the total cost of all trips. The authors consider a model where an offline sample of the trip sequence is available, obtained by including each trip independently with probability $\epsilon$.

The authors first show that a naive approach, which scales the sample by a factor of $1/\epsilon$ and applies the offline optimal strategy for the sample to the original instance, can have an arbitrarily large competitive ratio for the Bahncard problem, even though it is near-optimal for the ski rental problem. They then propose a more sophisticated algorithm and prove that its competitive ratio approaches $1.5$ as the ratio between the maximum trip cost and the card price goes to zero, improving over the optimal deterministic competitive ratio of $2$. They also propose a second algorithm and show that, under the additional assumption that trip costs are bounded away from zero, its competitive ratio approaches $1$. Finally, they support their theoretical results with experiments across different arrival patterns, price distributions, and sampling regimes.

**Compliance With Llm Reviewing Policy:**

Affirmed.

**Final Justification:**

My final recommendation is accept. I did not have any pressing concerns about the paper and the rebuttal did not change my evaluation.

**Key Questions For Authors:**

I do not have any questions for the authors.

**Limitations:**

yes

**Strengths And Weaknesses:**

**Soundness**

There are not many proofs in the main body and I did not verify the proofs in the appendix. That said, the problem setup, algorithms, and analysis appear technically sound.

The naive algorithm, which works well for the ski rental problem, is clearly explained and analyzed. The Bahncard instance where this approach fails provides good motivation for the more sophisticated algorithms. The algorithms themselves are described clearly and the paper provides helpful intuition throughout, which makes the reasoning easy to follow.

The experimental setup is described in detail and the results support the theoretical claims. One potential concern is that the competitive ratios depend on the ratio of the maximum trip cost to the card price approaching zero, and it is not clear how realistic this assumption is.

**Presentation**

The paper is well written and easy to follow. The problem and model are clearly defined, and the algorithms are described clearly.

The authors position their work well relative to prior literature. The overall story is logical: the failure of the naive algorithm motivates the improved algorithms, and the accompanying intuition helps guide the reader.

**Significance**

The paper studies a classic online problem in the framework of learning-augmented algorithms and shows that access to a simple random sample can significantly improve the competitive ratio.

The prediction model is very simple, which makes the approach potentially more practical than methods that rely on complex predictions. However, the paper does not discuss real-world applications of the problem and all experiments are conducted on synthetic data.

**Originality**

The sampling model itself is not new, but applying it to the Bahncard problem is a new direction. The paper shows that even very simple predictive information, such as a sparse random sample of the input, can lead to significant improvements in competitive ratio.

---

> ### Author Rebuttal · Authors · 2026-03-30
>
> We thank the reviewer for the detailed and positive evaluation, and for appreciating our work.

---

> > ### Author Rebuttal · Reviewer_kzr2 · 2026-04-02
> >
> > I will maintain my score.

---

> > > ### Author Response · Authors · 2026-04-04
> > >
> > > Thanks again for your supportive and encouraging evaluation of our work. We truly appreciate it.

---

### Official Review · Reviewer_VPYc · 2026-03-11

**Soundness:** 3
**Presentation:** 4
**Significance:** 2
**Originality:** 2
**Overall Recommendation:** 4
**Confidence:** 4

**Summary:**

This paper studies the Bahncard problem in a learning-augmented setting where the algorithm has access to information about a random sample of the input trips (which are revealed online). The paper explores how the competitive ratio can be improved beyond the classical 2 ratio, which is optimal for deterministic algorithms without predictions. The first algorithm presented achieves a competitive ratio approaching 3/2 as the ratio c (maximum trip cost to card cost) approaches 0. Algorithm 5.3 further improves this to approach 1 under the additional assumption that all trip costs lie in $[\alpha c ,c]$ for some constant $\alpha > 0$. The paper includes both theoretical analysis and experimental validation on various natural input distributions.

**Compliance With Llm Reviewing Policy:**

Affirmed.

**Final Justification:**

The authors addressed my questions on studying critical regimes and explained the difficulty of exploring more general settings. After consideration, I lean slightly towards acceptance, but I believe that the setting studied in the paper is limited and does not address the core question of learning-augmented algorithms, which is how to leverage potentially inaccurate predictions.

**Key Questions For Authors:**

1. Can we derive results depending on $\epsilon$? For example, assume that both $(c,\epsilon) \to 0$, then what is the limit CR depending on the relation between $c$ and $\epsilon$?
2. Can the authors report standard deviation on the figures, and add an experiment where the gap between the two algorithms is significant?
3. Have the authors tried proving lower bounds for the studied problem? If yes, what are the major technical difficulties why such bounds were not proved in the paper?
4. A setting closer to the standard learning augmented framework consists in assuming that the decision-maker only has access to predictions on the sampled trips, instead of having ground truth information on them. Such setting has been studied for Scheduling for example in "Non-clairvoyant scheduling with partial predictions". How difficult is it to generalize the current results to that setting?

**Limitations:**

yes

**Strengths And Weaknesses:**

### Strengths
- While the setting deviates from the standard learning augmented framework, considering access to samples of the inputs makes sense and is interesting. this work can inspire similar future research on other online algorithms.
- The paper is well structured and nicely builds intuitions on the problem and the algorithms. For example, the authors study first the ski-rental problem, which is a simpler version of the Bahncard problem. Also, they explain and illustrate why the naive algorithm can fail arbitrarily badly and explain its limitations.
- The paper has technical depth: the proposed algorithms are interesting, and the proofs are rigorous and non-trivial.

### Weaknesses
- The main theorems state the bounds as functions of c only, not explicitly showing the impact of the sample fraction size $\epsilon$ (hidden in $\phi$). The reader needs to go through the appendix to understand the dependency w.r.t $\epsilon$
- Similarly to the first question, studying the limit of CR when $c \to 0$ while considering $\epsilon$ constant doesn't give a clear understanding of the impact of $\epsilon$. For example if $\epsilon$ also converges to 0, how fast should $c$ converge to 0 relatively to $\epsilon$ for the results to hold?
- The paper doesn't prove any lower bounds, hence it is not clear if the algorithms are optimal (except in special regimes where CR $\to 1$)
- The paper only studies deterministic algorithms, and does not provide any results for randomized algorithms
- While Algorithm 5.3 has stronger bounds than Algorithm 5.2 (under assumptions), both algorithms seem to have the same performance in the experiments. The authors should include at least one experiment showing when the gap between the two occurs, or explain in more detail why this gap is unobserved.
- The figures only show the mean evolution, but do not report any uncertainty measures (for e.g. std)
- Typo: In the captions and legends of the figures, the algorithms are referenced as 4.2 and 4.3 instead of 5.2 and 5.3.

---

> ### Author Rebuttal · Authors · 2026-03-30
>
> We thank the reviewer for their insightful comments and address each concern below.
>
> Weakness 1
>
> The performance guarantees in their exact form are stated in the appendix. Due to the length and complexity of the expressions, they are abbreviated to asymptotic forms in the main body of the paper for easier comprehension. We will add the precise forms of the expression to the main body of the paper in the final version, as suggested by the reviewer.
>
> Weakness 2/Question 1
>
> The relative convergence of $\varepsilon$ compared to $c$ required for our algorithm can be directly computed from the exact forms of $\phi$ presented in the appendix. More precisely, one can derive that the convergence of the competitive ratio holds as long as $c/\varepsilon^2$ goes to $0$.
>
> Weakness 3/Weakness 4/Question 3
>
> We agree that understanding optimality is an important direction. For our results that approach a competitive ratio of $1$, the competitive ratio is clearly tight, but the rate of convergence and the assumptions on the input instance are potential directions for proving lower bounds. Proving general lower bounds in this setting is technically challenging due to the interplay between randomness in the sample and the online decision-making process, which breaks standard adversarial constructions. We view the question of obtaining lower bounds as an interesting direction for future work.
>
> Similarly, extending to randomized algorithms would require additional technical insights. Since there is already intrinsic randomness from the sampling, it is not immediately clear how further adding randomness in the algorithm can help in terms of its performance. In other words, an algorithm can extract randomness from the sample instead of generating new randomness, therefore allowing a randomized algorithm to be simulated by a deterministic one. Whether this simulation would be lossless, or quantifying the precise loss in the algorithm’s performance because of limitations of the amount of independent random bits that can be extracted from the sample, is an interesting and technically challenging direction for future work.
>
> We will include a discussion on lower bounds in the final version of the paper.
>
> Weakness 5/Question 2
>
> On cluster instances, the naive algorithm outperforms all other algorithms across different discount factors. For these families of instances, if we make the trips denser but less expensive, Algorithm 5.3 will have performance closer to the naive algorithm and will therefore outperform Algorithm 5.2. However, reaching this regime requires extremely dense/large instances, which leads to memory limitations in simulation. We will clarify this in the final version of the paper.
>
> Weakness 6/Weakness 7/Question 2
>
> Thank you. We will include the standard deviations and fix the labeling typos in the final version of the paper.
>
> Question 4
>
> Our framework assumes accurate but sparse samples, emphasizing limited information rather than noisy predictions. We are aware of one paper (Zhao, Tang, Chen, and Deng, Neurips 24) that uses possibly noisy predictions for the Bahncard problem. However, this work does not achieve any improvement over the worst-case bounds if the discount factor $\beta = 0$. (Note that as mentioned in our paper, $\beta = 0$ is the most challenging setting, since the algorithm’s choices determine the largest share of the cost in this case.) In contrast, we show that a sparse sample of the input can improve over worst-case performance, even for the case of $\beta = 0$ which subsumes all other values of $\beta$.
>
> We also note that we empirically evaluate the performance of our algorithms in a noisy setting where the sample can have errors (see Section 6 and Appendix E). More precisely, we first add noise to the original instance, then sample from this noisy instance, and use these samples to guide the algorithm on the original instance. The noisy instance is created from the original instance by randomly adding false negatives and false positives: we randomly delete some existing trips and add some non-existent ones. This methodology follows prior work by Zhao et al. in Neurips 24. The experimental results (see Figures 7-14 in Appendix E) demonstrate that our algorithms remain robust to sampling errors across several natural families of instances.
>
> We hope these clarifications address the reviewer’s concerns. We are happy to provide further details.

---

> > ### Author Rebuttal · Reviewer_VPYc · 2026-04-01
> >
> > I thank the authors for their detailed response!
> >
> > Regarding Weakness 1, I understand that including the complete formulas in the main body may affect readability. However, I suggest at least providing additional intuition about $\phi$ and offering more discussion of the critical regimes (for example, when the threshold $c/\varepsilon^2$ approaches $0$) and relations between the parameters and their impact, so that readers can better understand and interpret the results.
> >
> > Regarding lower bounds and exploring random algorithms, I agree that these can be more technical and more difficult to study.
> >
> > However, I still believe that studying potentially inaccurate predictions is essential in the field of learning-augmented algorithms. The remark raised by the authors in the rebuttal regarding the regime $\beta = 0$ highlights an important gap when transitioning from accurate (sparse) predictions to noisy ones. One would typically expect these two regimes to interpolate continuously as the prediction error increases from zero, which further shows the importance of investigating this question.
> >
> > I will raise my recommendation to "weak accept".

---

> > > ### Author Response · Authors · 2026-04-02
> > >
> > > Thank you for your thoughtful and constructive feedback on our rebuttal. We greatly appreciate your perceptive and positive comments.
> > >
> > > We will include a discussion clarifying how the parameters interact within the full expression of the performance guarantee.
> > >
> > > We also agree that understanding noise sensitivity is an important direction. Since prior results do not yield improvements over worst-case bounds in the full discount setting ($\beta = 0$), which is arguably the most important case since it subsumes all other values of $\beta$, the first step was to obtain a learning-augmented result for this setting. Now that we know that a sample can improve performance bounds even for $\beta = 0$, the natural next step would to be to understand whether a noisy sample suffices.

---

### Official Review · Reviewer_9snV · 2026-03-12

**Soundness:** 2
**Presentation:** 3
**Significance:** 2
**Originality:** 2
**Overall Recommendation:** 4
**Confidence:** 3

**Summary:**

This paper studies the Bahncard problem in a learning-augmented setting where the algorithm receives a random sample of future trips. The authors first show that a natural baseline that optimizes over the sampled instance can perform arbitrarily poorly due to sampling gaps. To address this issue, they propose an algorithm that corrects such gaps by merging nearby card purchases inferred from the sample. They prove that the resulting algorithm achieves a competitive ratio approaching (3/2) as the maximum trip cost (c) becomes small, improving upon the classical deterministic bound of (2). Experimental results further illustrate the empirical performance of the proposed approach.

**Compliance With Llm Reviewing Policy:**

Affirmed.

**Final Justification:**

The authors have adequately addressed my questions and concerns. However, as I still have some reservations regarding the model's assumptions, I am inclined to give a weakly positive score.

**Key Questions For Authors:**

1.Although the introduction briefly discusses the motivation behind the problem formulation, it remains unclear how realistic the assumed information model is in practice (see Weakness 1). In particular, the algorithm assumes access to perfectly accurate samples drawn from the future instance. Could the authors provide a more detailed discussion of the practical motivation for this model and clarify in what real-world scenarios such information might be available?

2.The proposed algorithm relies on filling short gaps inferred from the sampled solution. Could the authors provide additional intuition on why the chosen threshold  is appropriate? In particular, is this threshold optimal for the analysis, or could alternative gap-handling strategies potentially yield improved theoretical guarantees?

3.The sampling model assumes that each trip is independently sampled with probability . Could the authors comment on how sensitive the algorithm and the theoretical guarantees are to deviations from this assumption? For example, how robust is the algorithm if the sampling process is biased, correlated, or otherwise differs from the independent sampling model assumed in the analysis?

**Limitations:**

yes

**Strengths And Weaknesses:**

Strengths:

1.The idea of leveraging sampled information to improve the classical competitive ratio bounds for the Bahncard problem is interesting.

2.The paper is generally well written and clearly organized. The presentation of the problem setting and the algorithms is relatively easy to follow.

Weaknesses:

1.From the perspective of the problem formulation, the information model studied in this paper appears to be closer to a missing-data or random sampling model rather than a prediction-based advice model typically considered in learning-augmented algorithms. In particular, the sample is drawn directly from the future instance and is assumed to be perfectly accurate. Such information may be difficult to obtain in many realistic scenarios, which raises questions about the practical relevance of the model.

2.The theoretical performance metric is defined as the expected competitive ratio with respect to the sampled information. Since the final performance guarantee aggregates the effect of multiple sampled observations, it is unclear how informative the guarantee is when only a single sample is available.
 In fact, if sufficiently many accurate samples are provided, the algorithm may essentially reconstruct most of the trip sequence, making the problem significantly easier. As a result, it is unclear whether the reported guarantees truly reflect the difficulty of the original online problem, rather than the advantage gained from having extensive sampled information.

3.The main theoretical result achieving a competitive ratio approaching  relies on the assumption that , i.e., that the individual trip costs are sufficiently small and trips are sufficiently dense. This condition appears to be quite restrictive, and it is unclear how often it holds in realistic settings. Although Section 7 briefly discusses this limitation, it would be valuable to provide a competitive ratio guarantee that does not depend on the asymptotic regime , even if the bound is slightly weaker, in order to obtain a more robust theoretical guarantee.

---

> ### Author Rebuttal · Authors · 2026-03-30
>
> We thank the reviewer for their insightful comments and address each concern below.
>
> Weakness 1/Question 1
>
> Our model follows prior work on online algorithms in the Online Algorithm with a Sample framework (Section 2). This model was proposed under the umbrella of learning-augmented algorithms as an alternative to algorithms with predictions by Argue, Frieze, Gupta, and Seiler (NeurIPS 22). Note that in the algorithms with predictions framework for the Bahncard problem, to approach the desired consistency guarantee, the predictor needs to produce near-perfect predictions for almost every trip. In contrast, in our algorithm, we only require a very sparse sample to obtain an improvement over the classical online algorithm.
>
> Another line of work on learning-augmented algorithms for the Bahncard problem (Zhao, Tang, Chen, and Deng, NeurIPS 24) uses short-term predictions. But this work does not achieve any improvement over the worst-case bounds if the discount factor $\beta = 0$. (As mentioned in our paper, $\beta = 0$ is the most challenging setting, since the algorithm’s choices determine most of the cost.) In contrast, we show that a sparse sample of the input can improve over worst-case performance, even for $\beta = 0$, which subsumes all other values of $\beta$.
>
> We also empirically evaluate the performance of our algorithms in a noisy setting where the sample can have errors (see Section 6 and Appendix E). More precisely, we first add noise to the original instance, then sample from this noisy instance, and use these samples to guide the algorithm on the original instance. The noisy instance is created by randomly adding false negatives and false positives: we delete some existing trips and add some non-existent ones. This methodology follows Zhao et al. (NeurIPS 24). The experimental results (Figures 7–14 in Appendix E) demonstrate that our algorithms remain robust to sampling errors across several natural families of instances.
>
> Weakness 2
>
> In our model, a single sample already contains multiple trips, since each trip is included independently with probability $\varepsilon$. Providing multiple independent samples is equivalent to increasing the effective sampling rate and therefore does not fundamentally change the model. Our analysis applies to any sampling rate $\varepsilon$, including small values where only limited information is available.
>
> Importantly, even with access to the sample, reconstructing the original instance remains highly nontrivial. The most natural approach, rescaling the sampled trips by $1/\varepsilon$, can perform arbitrarily poorly, as we show in Section 5.1. This demonstrates that the algorithm cannot simply recover the full instance from the sample, and that the problem retains its intrinsic online difficulty. We overcome this by algorithmic modifications to achieve a near-optimal solution, as described in the paper.
>
> Weakness 3
>
> The performance guarantees in their exact form are stated in the appendix. Due to the length and complexity of the expressions, they are abbreviated to asymptotic forms in the main body for easier comprehension. We will add the precise expressions to the main body in the final version, as suggested.
>
> Question 2
>
> The intuition of Algorithm 5.2 comes from the adversarial instance in Section 5.1. The naive reconstruction-based algorithm performs poorly due to short gaps with dense trips. We eliminate short gaps by buying additional cards, which improves the performance guarantee.
>
> We assume the chosen threshold refers to $\delta$. The value $\delta = c^{1/5}$ is near-optimal. From the exact bounds, one can derive optimal values of $\delta$, but these have a convoluted form and only improve lower-order terms. Thus, we use $\delta = c^{1/5}$ for clarity without affecting the main guarantee.
>
> Question 3
>
> Independence of sampling is an important requirement for results in the Online Algorithms with a Sample framework. If samples can be manipulated by the adversary, they may reveal only unimportant parts of the input, providing no advantage. In our problem, for every contiguous sequence of trips, we require concentration bounds on the sample, which generally do not hold without independence. Since this is the only place where independence is used, it suffices to have any sampling process where these concentration inequalities hold. However, allowing arbitrary correlation even within one contiguous group can make the sample useless.
>
> We hope these clarifications address the reviewer’s concerns and are happy to provide further details.

---

> > ### Author Rebuttal · Reviewer_9snV · 2026-04-03
> >
> > Thank you for the detailed response. However, I still have reservations regarding the method's reliance on accurate predictions and certain underlying assumptions. I will adjust my score accordingly.

---

> > > ### Author Response · Authors · 2026-04-04
> > >
> > > Thank you for reading our rebuttal comments carefully and for your positive and perceptive response.
> > >
> > > We agree that noise sensitivity is an important direction. However, we would like to emphasize that our work is the first to establish a learning-augmented improvement across all values of $\beta$, and in particular for the case of full discount ($\beta = 0$), which is both natural and arguably the most important, since it subsumes all other values of $\beta$. Notably, prior work on the learning-augmented Bahncard problem (Zhao et al., NeurIPS ’24) degenerates to worst-case guarantees when $\beta = 0$.
> > >
> > > Thus, our results advance the state of the art by demonstrating that a learning-augmented approach can provably improve performance for the Bahncard problem in general, including in the most challenging regime of $\beta = 0$. In this context, investigating robustness to noise is a natural and important next step, building directly on our contribution; we also provide empirical results in this direction through numerical simulations.
> > >
> > > We hope this clarifies the novelty and significance of our results. We would be happy to address any further questions.

---

### Decision · Program_Chairs · 2026-04-30

**Decision:**

Accept (regular)

**Comment:**

This paper proposes an algorithm for the classic Bahncard problem in the setting where the algorithm can see a sample of the upcoming trips. This fits in general the topic of learning-augmented algorithms or algorithms with predictions, but in this case the choice of prediction is very natural. This is not the first paper to propose this choice (though it's not a common choice either), but the first one to apply it to this problem.

The reviewers agree that the idea is interesting, and that the paper is very well written. Some reviewers mention that there is also some technical depth in arguments used in the paper.

The main weakness mentioned by two reviewers is that the results hold in the limit, or in other words, it is assumed that certain parameters are sufficiently small. This does not seem to be a big limitation though.

In general, this looks like a clear accept.